

# Extrapolation is not enough: Impacts of extreme land-use change on wind profiles and wind energy according to regional climate models

Jan Wohland[1], Peter Hoffmann[2], Daniela C.A. Lima[3], Marcus Breil[4], Olivier Asselin[5], and Diana Rechid[2]

[1]Institute for Atmospheric and Climate Science, ETH Zurich, Zurich, Switzerland
[2]Climate Service Center Germany (GERICS), Helmholtz-Zentrum Hereon, Hamburg, Germany
[3]Universidade de Lisboa, Faculdade de Ciências, Instituto Dom Luiz
[4]Institute of Physics and Meteorology, University of Hohenheim, Stuttgart, Germany
[5]Ouranos, Montréal, QC H3A 1B9, Canada

**Correspondence:** Jan Wohland (jan.wohland@env.ethz.ch)

**Abstract.** Humans change the climate in many ways. In addition to greenhouse gases, climate model inputs thus include a number of other forcings like land-use change. While studies typically investigate the joint effects of all forcings, we here isolate the impact of afforestation and deforestation on winds in the lowermost 350 m of the atmosphere to quantify the relevance of the lower boundary condition for large-scale wind energy assessments. We use vertically resolved sub-daily output

from two regional climate models instead of extrapolating near-surface winds with simplified profiles. Comparing two extreme scenarios, we report that afforestation reduces wind speeds by more than 1 m/s in many locations across Europe even 300 m above ground and thus remains relevant at hub heights of current and future wind turbines. We show that standard extrapolation with modified parameters approximates long-term means well but fails to capture essential spatio-temporal details, such as changes in the daily cycle, and is thus insufficient to estimate wind energy potentials. Using adjacent climate model levels

to account for spatio-temporal wind profile complexity, we report that wind energy capacity factors are strongly impacted by afforestation and deforestation: they differ by more than 0.1 in absolute terms and up to 50% in relative terms. Our results confirm earlier studies that land use change impacts on wind energy can be severe and that they are generally misrepresented with common extrapolation techniques.





## 1 Introduction

The last two decades have seen large progress in the deployment of renewable energy, in terms of technological advances as well as reduced costs (IRENA, 2022). However, ambitious climate change mitigation requires further deep and rapid emission reductions ultimately reaching net-zero or net-negative emissions (IPCC, 2022) which many countries aim to achieve by adding substantial amounts of new renewable capacity. Ambitious goals are widespread across Europe: The European Union aims at carbon neutrality in 2050 (Commission, 2021), Switzerland targets climate neutrality by 2050 (Confederation, 2022), and the

United Kingdom aims at net-zero emissions by 2050 (UK, 2008). The European power system will therefore dominantly rely on weather dependent power generation from wind and solar energy in just a few years. This reliance means that we need to better understand renewable resources, including potential changes caused by human activity, to build power systems that function reliably.

Concern about adverse impacts of climate change on wind energy potentials has motivated a range of studies (e.g., Hueging

et al., 2013; Tobin et al., 2016; Reyers et al., 2016; Moemken et al., 2018; Karnauskas et al., 2018; Schlott et al., 2018; Soares et al., 2019; Lima et al., 2021; Hahmann et al., 2022). Drawing from those papers and other published peer-reviewed literature, a recent review (Pryor et al., 2020) and the sixth Assessment Report of the IPCC (IPCC, 2022) conclude that climate change will only weakly impact the wind resource and that it remains unclear in most locations whether winds will be amplified or weakened. However, there are many possible interpretations of this conclusion, including (a) that climate change will indeed

only have limited impacts on wind energy, (b) that climate models are deficient in capturing dynamical changes and therefore underestimate the real risks, (c) that methodological flaws in the state-of-the-art to convert climate model output to wind power generation hinder signal detection, and (d) that the effects of different aspects of human activity happen to cancel out.

It is worthwhile to consider option (b) because strong reliance on structurally similar models can be problematic if inter-model agreement is mistakenly interpreted as evidence for reliability (e.g., Thompson, 2022). Current climate models are

structurally similar. On the one hand that is because they all approximate the same climate system obeying the same physical laws, providing a good reason for structural similarity. On the other hand, coupled climate models also share sub-components such as the same atmospheric or ocean model or are derived from a common older predecessors (Knutti et al., 2013), which introduces structural similarity that has practical justification but lacks physical justification. More importantly, however, current models have documented deficiencies in capturing important aspects of atmospheric dynamics. For instance, the storm

track is generally positioned too close to the equator and is oriented too much along the East-West direction, which might be an artefact of the current resolution in global climate models (Schemm, 2023). And observed late winter jet stream variability on multidecadal scales is not realistically captured in current climate models, likely due to issues in the ocean-atmosphere coupling (Simpson et al., 2018). Moreover, climate models underestimate the importance of multidecadal variability in the mid-latitudes (O'Reilly et al., 2021), potentially implying that important processes are not well captured and calling into question

the fitness-for-purpose of current climate models for wind energy assessments.

Given these issues, it makes sense to unpack the problem step-by-step. In this study, we therefore ignore dynamical changes caused by altered greenhouse gas and aerosol concentrations and instead explore options (c) and (d). A potential methodological



flaw is the conversion of winds from near-surface to hub height using extrapolation techniques. Indeed, Soares et al. (2020)
report that the extrapolation introduces sizeable errors of up to 30% during the historical period compared to using model level
information from ERA5. Nevertheless, most of the above cited climate change literature extrapolates with a wind profile that
even remains constant over the 21st century, with two exceptions. First, Hahmann et al. (2022) derive North Sea hub height
winds from model levels and report that the power law performs poorly in comparison. This result is striking because unlike
over land, the lower boundary condition of the North Sea does not change directly as a function of the prescribed scenario
but only indirectly via changes in sea state. Over land, however, the Climate Model Intercomparison Project (CMIP) scenarios
include substantial changes in land-use during the 21st century (Hurtt et al., 2011, 2020), suggesting that onshore wind profiles
may change more drastically and the power law is even less well suited. Second, Wohland (2022) restricted his analysis to
surface winds and consequently did not require a conversion to hub height. However, he found large-scale systematic difference
in future winds projected by regional climate models from EURO-CORDEX (Jacob et al., 2014) and the driving global climate
models from CMIP5 (Taylor et al., 2012) that are caused by land use change.

The effects of land use change and dynamical changes on winds indeed cancel out in some scenarios. Analysing the MPI
Grand Ensemble, and estimating the dynamical change from idealized simulations without land use change, Wohland et al.
(2021b) show that global mean onshore wind speed would drop by about -0.25 m/s due to dynamical reasons while it would
increase by about 0.25 m/s because of land use change, yielding zero change in total. These numbers correspond to the rcp85
scenario and compare the end of the 21st century with the beginning of the historical period. It remains unclear, however,
how this balance of effects evolves with height, although it appears plausible that the dynamical effects would "win" since the
land-use effects are expected to decay with height.

In this study, we isolate the effect of afforestation and deforestation on wind energy potentials. We do so by analysing high-
resolution regional climate model output that allows to compute hub height wind speeds without reliance on constant parameter
extrapolations, thereby tackling a potential methodological flaw in the published literature. Moreover, we use experiments that
only differ in the lower boundary condition which allows to investigate the effect of land use independently from the effects
of other climate forcings, thereby enabling better understanding of the different effects that may mask each other in combined
scenarios.

## 2  Data and Methods

We use data from the coordinated regional climate model inter-comparison project "Land Use and Climate Across Scales"
(LUCAS, see Davin et al., 2020). LUCAS is a World Climate Research Programme (WCRP) flagship pilot study that inves-
tigates the biogeophysical effects of land use changes in Europe (e.g., Breil et al., 2020; Sofiadis et al., 2022; Mooney et al.,
2022; Daloz et al., 2022). Within LUCAS, experiments with extreme land-use change scenarios were conducted from which
we quantify the maximum effects that land-use change may have on future wind energy resources. In the GRASS scenario, all
suitable areas are covered by grass, whereas in the FOREST scenario, all suitable areas are covered by forests.



80 We report wind speed differences between the GRASS and FOREST simulations as

$$\Delta s = s_{\text{GRASS}} - s_{\text{FOREST}}. \tag{1}$$

That is, positive differences correspond to stronger winds in the GRASS scenario as compared to the FOREST scenario. We refer to $\Delta s$ as the effect of deforestation. Since the effect of deforestation is the opposite effect of afforestation and both share the same absolute values, we use the terms interchangeably when discussing the magnitude of change.

## 2.1 Model subset

To assess the effect of deforestation and afforestation on wind energy, we need temporally highly resolved model output because wind energy potential depends non-linearly on wind speeds, and we need vertically highly resolved model output because wind turbines roughly populate the lowermost 300m of the atmosphere. In particular, we use models that

1. provide sub-daily winds

2. make model level information available

3. have at least three atmospheric model levels in the lowermost 350 m

and we discard the rest.

Based on these criteria, we identify two suitable models out of 6 models that were provided by LUCAS member groups. The suitable models are REMO-iMOVE run by the Climate Service Center Germany (GERICS) and WRFaNoahMP run by Institute Dom Luiz (IDL), see Table 1 for an overview. Throughout the manuscript, we refer to the model output by naming the institution (i.e., GERICS or IDL). We acknowledge that an ensemble size of two is limited, however, it is the best currently available data base for this analysis. Even with only two models, the results of this study are relevant because we are the first to systematically investigate land-use driven changes in winds, wind profiles, and wind energy potentials using regional climate models.

## 2.2 Resolution

Both models have a horizontal resolution of 0.44° covering Europe on a grid with a rotated pole, following the standard CORDEX domain definitions (see EUR-44 in CORDEX, 2015). This resolution corresponds to about 50km. Temporal resolution is sub-daily for both models, and IDL provides hourly output while GERICS only outputs every 6 hours.

We use the lowermost 3 (4) models levels for GERICS (IDL), corresponding to approximate heights at 30m, 140m, and 340m (28m, 95m, 190m, 300m). The approximate heights are computed by subtracting orography from the geopotential height fields averaged over the entire domain and a full year. The exact height above ground of a model level varies in time and space due to the use of terrain-following hybrid vertical coordinates and we take these variations into account in the wind power conversion (see Sec. 2.5). We ensured, however, that the approximate heights can be used as a reasonable proxy for the exact heights in the context of long-term changes because the variations are relatively small. For example, the lowest GERICS model





level sits 28.1m above ground on average. The temporal standard deviation averaged over all locations is 0.8 m and the spatial standard deviation averaged over all timesteps is 0.7 m. That is, the variations of the level height are at the order of a few per cent, as expected, since they originate from atmospheric pressure and density differences which are also small compared to their means. The total number of vertical levels is 50 in the IDL simulations and substantially smaller in the GERICS simulations (27) (see Table 1 in Davin et al. (2020) for additional details).

**2.3    Area and period of interest**

To compute onshore winds, we use data over the European continent extending east up to approximately Ankara, Turkey (see Fig. S10 for details). We only use grid boxes that are contained in both data sets. Since the GERICS simulations contain a larger sponge zone, that means that we only use 106 x 103 of the available 129 x 121 grid boxes.

    We compute mean wind speeds by averaging over model years 1986 to end of 2015 as suggested by the LUCAS consortium,

addressing changes in the annual means as well as the seasonal and daily cycles. When comparing the daily cycles, we follow a downsampling rather than an averaging approach: to compare the GERICS output (0AM, 6AM, 12AM, 6PM; all times UTC) with the IDL output (0AM, 1AM, 2AM etc.; all times UTC), we only consider time steps that are available in both datasets and ignore the additional information contained in IDL at the other hours. We argue that this approach is most meaningful because wind speeds are reported as instantaneous values and the choice of output frequency does not affect the output.

By contrast, we use data at the highest available frequency to calculate wind power, namely hourly in IDL and 6 hourly in GERICS. IDL uses modeling time steps of 90 seconds, while GERICS uses 240 seconds.

**2.4    Comparison to the log and power laws**

We do not rely on heuristics to extrapolate from near-surface winds to hub heights because we build on models that provide winds at multiple heights in the relevant domain. However, we compare the results obtained from the LUCAS models with

standard extrapolations to gain insights into the errors made by the extrapolation. Specifically, we compare against the log and power law which are dominantly used in the literature (e.g., Bloomfield et al., 2016; Tobin et al., 2016; Wohland et al., 2017; Schlott et al., 2018; Soares et al., 2019; van der Wiel et al., 2019; Pryor et al., 2020).

    The log law relates winds $s$ at heights $z$ and $z'$ and location $\boldsymbol{x}$ via the roughness length $z_0$ and the displacement height $d$ as

$$\frac{s(\boldsymbol{x}, z, t)}{s(\boldsymbol{x}, z', t)} = \frac{ln\left(\frac{z - d(\boldsymbol{x})}{z_0(\boldsymbol{x})}\right)}{ln\left(\frac{z' - d(\boldsymbol{x})}{z_0(\boldsymbol{x})}\right)}. \tag{2}$$

The power law only depends on the power law exponent $\alpha$:

$$\frac{s(\boldsymbol{x}, z, t)}{s(\boldsymbol{x}, z', t)} = \left(\frac{z}{z'}\right)^{\alpha} \tag{3}$$

    The log law can be formally derived as a special case of the general equations of motion under neutral stratification and the power law is empirically motivated (e.g., Emeis, 2013). We highlight the location and time dependency in Equations 2 and 3 to illustrate two assumptions of these approaches that limit their applicability. First, both approaches assume that the wind profile



**Table 1.** Overview of the models used. Approximate height are given for all model levels in the lowermost 350 m of the atmosphere. We refer to the model simulations by the name of the institution.

| Institution | Model | Horizontal Resolution | Temporal resolution | Approximate heights |
|---|---|---|---|---|
| GERICS | REMO-iMove | 0.44° (EUR-44) | 6 hourly | 30m, 140m, and 340m |
| IDL | WRFa-NoahMP | 0.44° (EUR-44) | 1 hourly | 28m, 95m, 190m, 300m |

is constant in time because the right hand side has no time dependency. Second, the power law generally does not include a spatial dependency of the wind profile, although some studies use different values for $\alpha$ over land and over ocean, to partially address this issue (e.g., Hueging et al., 2013).

When comparing model level results with extrapolations (in Fig. 3), we use the roughness lengths reported in Breil et al. (2020) for IDL, assuming a 50/50 split between coniferous and deciduous tress. The GERICS values from that paper, however, can not be directly used because the contribution from subgrid-scale orography is missing. We therefore computed onshore

mean effective roughness lengths from the climate model (FOREST: $z_o = 1.686$m, GRASS: $z_o = 0.693$m) and use those values.

### 2.5 Wind power conversion

Despite using the model level output, we need to perform some corrections as the model layers do not perfectly match the tur-

bine hub height. We interpolate rather than extrapolate, allowing for spatio-temporal variations of the wind profile. Specifically, we compute a temporally and spatially evolving coefficient $\alpha'(\boldsymbol{x}, t)$ based on Eq. 2 from the adjacent levels as

$$\alpha'(\boldsymbol{x}, t) = log_{\frac{z_{\text{above}}}{z_{\text{below}}}} \left( \frac{s(\boldsymbol{x}, z_{\text{above}}, t)}{s(\boldsymbol{x}, z_{\text{below}}, t)} \right), \tag{4}$$

where $z_{\text{above}}$ and $z_{\text{below}}$ are the height of the layer above and below hub height, respectively. Those heights are also a function of time and location but we don't highlight this dependency in the equation to ease legibility.

We then apply the obtained coefficient to interpolate to hub height using Eq. 2. In Supplementary Fig. S1, we show an illustration of the approach for a few locations, highlighting that $\alpha'$ is far from constant and instead varies vastly in the range from about -0.2 to 0.6. This range includes the widely used value of $1/7 \approx 0.14$. However, the large scatter around that value indicates that a constant value does capture the complexity of wind profiles.

In a last step, we multiply hub height wind speeds with the power curve of a SWT120-3600 turbine to obtain capacity factor

(CF) timeseries. The power curve is taken from the Windpowerlib database (Haas et al., 2019). This turbine has a hub height of 120 meters and was chosen following Wohland et al. (2021a) as it represents the median current wind turbine.





## 3  Results and Discussion

Our core finding is that afforestation/deforestation matters for winds around hub height according to both models. For instance, wind speeds drop by more than 1 m/s even 340 m above ground in large parts of Poland, Ireland and Belarus in the GERICS simulations (Fig. 1i), and they drop by at least 0.5 m/s virtually everywhere else on the continent. In fact, wind speed drops greater than 1 m/s occur in almost half of the onshore grid cells at 30m and in 10% of grid cells at 340m above ground (Table 2). IDL features reductions of similar magnitude, even exceeding 1.5 m/s in large parts of Scandinavia and Iceland near the surface that slowly decay with height (Fig. 2l). About 25% of onshore grid cells still experiences reductions greater than 1 m/s at 190m above ground, dropping to 14% at 300m (Table 2).

Interestingly, the models also agree that deforestation has non-local effects on wind speeds, namely by impacting offshore winds. While a clear land-sea divide exist in both models and at all levels (Fig. 1c and 2c), mean offshore winds decrease by 0.34 m/s at 190m and by 0.26 m/s at 140m according to IDL and GERICS, respectively (Table 2).

While models agree on the change amplitudes, they disagree considerably about the location of maximum wind speeds. For example, GERICS positions the offshore wind speed maximum North West of the UK, and IDL locates it much further East. The models also disagree with respect to the location of maximum wind changes. While GERICS winds decline strongest in a band ranging from the UK to Poland via Benelux and Germany, IDL positions the center of change around Scandinavia, documenting significant model uncertainty. This uncertainty is a combination of at least two factors, namely (a) how the models (or modelers) translate the scenario into actual boundary conditions and (b) how the models respond to those boundary conditions. For instance, in the FOREST scenario, forests grow in all suitable locations but models can disagree as to whether an area is suitable and whether to include any subgrid-scale information. For example, the GERICS simulations follow a tile approach that allows for combinations of different land uses within a grid box whereas IDL adopts a majority approach where each grid box has exactly one land use type (Davin et al., 2020). The inter-model difference manifests, for example, in the lowest model level over the Alps where the signal in GERICS is much weaker than in IDL (cf. Figs. 1c and 2c). This is because alpine land cover changes are small in the GERICS simulations due to the high bare ground fraction. Additionally, disagreement can stem from the allocation of land surface surface parameters (e.g., roughnes length, leaf area index) and whether and how those parameters evolve throughout the year.

To test the robustness of change in different parts of the year, we analyzed the changes per season (see Supplementary section 4). Overall, we find qualitatively similar changes in all seasons but the amplitude of change is modulated. In particular, the summer signal (June-July-August) is weaker than the winter signal (December-January-February).

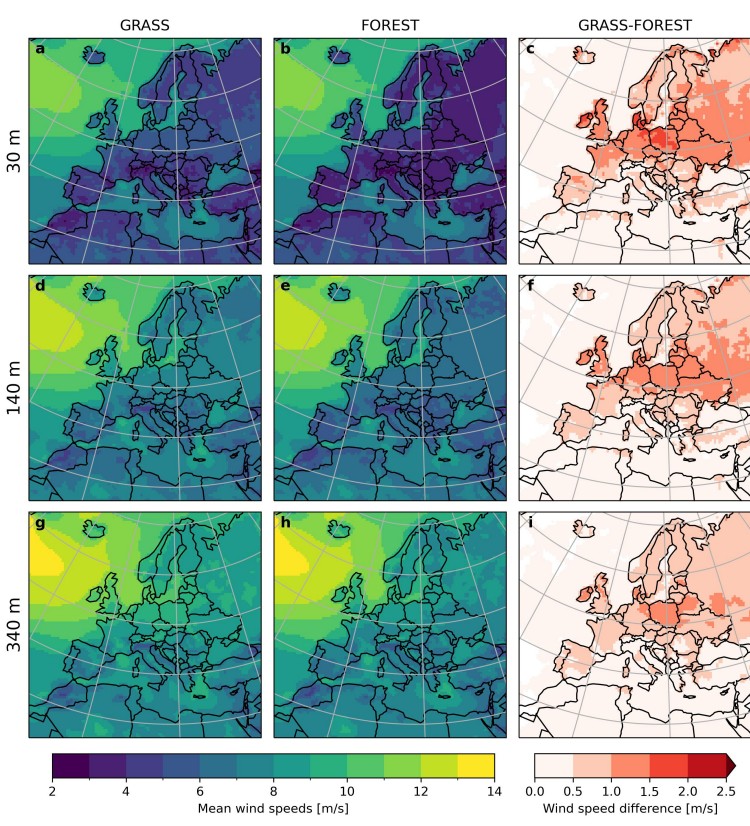

**Figure 1.** Mean wind speeds in the GRASS (a, d, g) and FOREST (b, e, h) scenario, as well as their difference (c, f, i), based on the GERICS simulations. Data is plotted for the lowermost 3 model levels which have approximate heights of 30m, 140m and 340m.





**Table 2.** Overview of mean wind speed changes between the FOREST and GRASS simulation. Values correspond to the data shown in Figs. 1 and 2. Onshore fraction refers to the fraction of land grid cells that experience a wind speed change greater than a threshold of 0.5 m/s or 1m/s, respectively.

| Institution | Approximate height | Mean onshore change [m/s] | Onshore fraction > 0.5 m/s [%] | Onshore fraction > 1 m/s [%] | Mean offshore change [m/s] |
|---|---|---|---|---|---|
| GERICS | 30 m | 0.95 | 91 | 45 | 0.28 |
| | 140 m | 0.82 | 87 | 28 | 0.26 |
| | 340 m | 0.71 | 80 | 10 | 0.23 |
| IDL | 28 m | 1.24 | 97 | 68 | 0.41 |
| | 95 m | 0.96 | 95 | 31 | 0.37 |
| | 190 m | 0.81 | 86 | 23 | 0.34 |
| | 300 m | 0.69 | 71 | 14 | 0.32 |



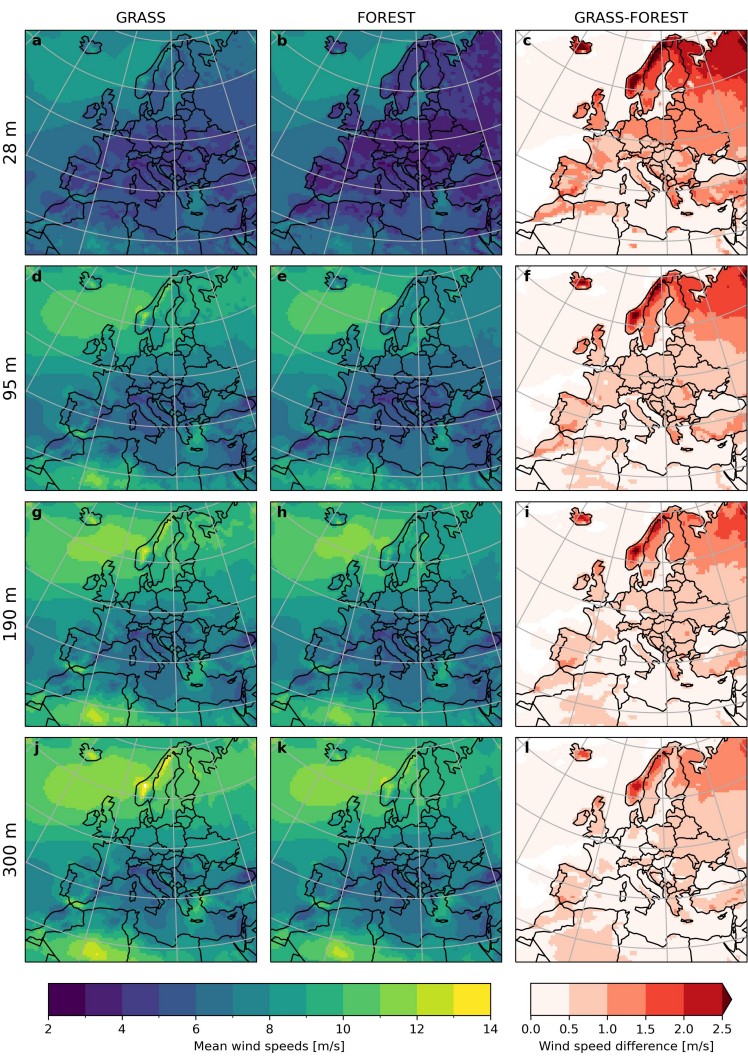

**Figure 2.** Mean wind speeds in the GRASS and FOREST scenario, as well as their difference, based on the IDL simulations. Data is plotted for the lowermost 4 model levels which have approximate heights of 28m, 95m, 190m and 300m.





## 3.1 Onshore mean signal decay with height

Given that extrapolation from 10m winds is the most common approach in wind energy related climate research, we now address the discrepancy between extrapolation and explicit modeling. Fig. 3 shows the evolution of a near-surface wind speed change with height. It documents qualitatively different results, where some approaches show that the surface perturbation grows with height while others show the opposite.

Surface change grows with height when using the power or log law with the same profile in the GRASS and FOREST scenarios (i.e., following Eq. 2 and 3 with constant $\alpha = 1/7$, $z_0 = 0.05$ m corresponding to grass, and $d = 3$ m). The reason is simple: With the same profile, the near surface winds in both scenarios are scaled with the same monotonically increasing profile, and it follows that the difference between the two also increases monotonically with height.

By contrast, the model output suggests that the surface change decays with height, in line with physical expectation. The seeming contradiction between extrapolation and model output can be reconciled by using modified extrapolation parameters for the GRASS and FOREST experiments. Using the model specific roughness lengths (see Sec. 2.4), we find good agreement between the explicitly modeled decay and the extrapolated decay.

These results have practical implications because CMIP scenarios for the 21st century contain substantial land use change. Our results mean that land use change impacts on hub height winds are exaggerated when using the same scaling with different lower boundary conditions. For instance, 10m wind speeds are projected to decrease by up to 0.5 m/s in Eastern Europe in the RCP4.5 climate change scenario by the end of the 21st century due to changes in the lower boundary condition (Wohland, 2022). Following Fig. 3, this near-surface reduction is amplified by about 40% to 0.7m/s at 100m above ground using the unmodified log or power law. By contrast, it is reduced by about a third according to the GERICS and IDL simulations to 0.35m/s. In other words, ignoring the change in the wind profile leads to an overestimation by a factor of two at hub height. Since dynamical changes, for example, from modified storm tracks or altered temperature gradients, are not scaled in a similar fashion, it follows that the detection of climate change signals for hub height winds might be hampered, as the relative role of dynamical changes is underestimated.

Nevertheless, the results show that the decay modeled by GERICS and IDL is fairly linear in log space and can thus be approximated well by the log and power law *on average over all seasons, 30 years, and an entire continent*. We will explore in the rest of the paper whether the modified extrapolations are also suitable at specific locations and seasons, and whether it correctly modifies quantities that depend non-linearly on wind speeds.





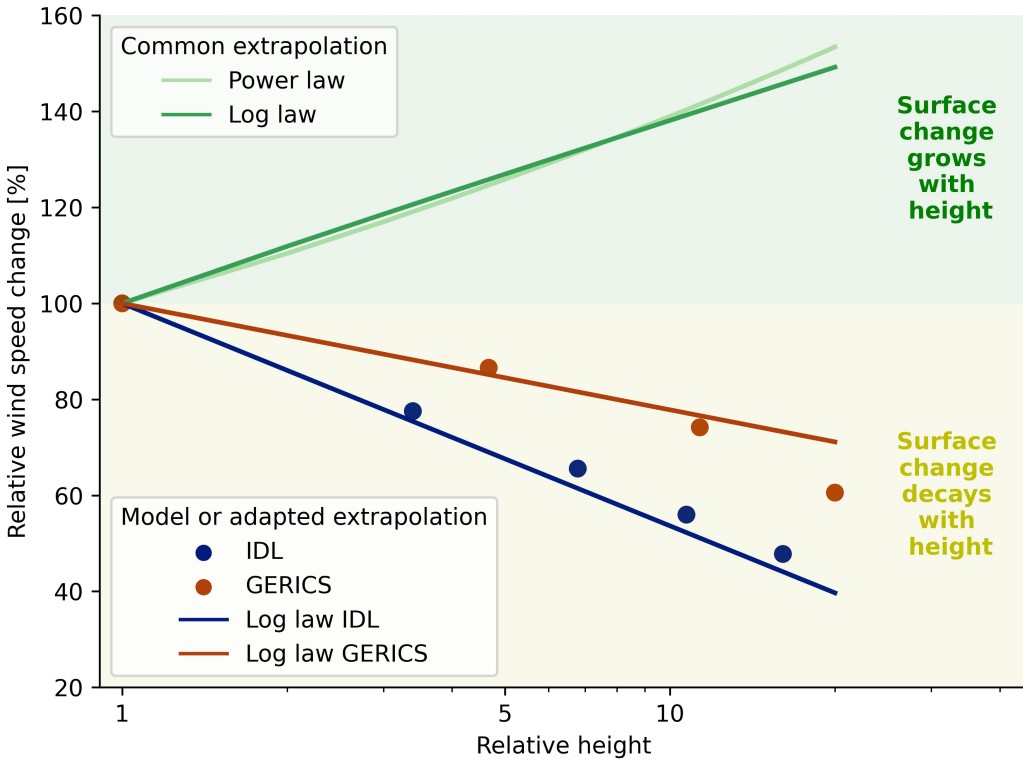

**Figure 3.** Evolution of relative wind speed change with height according to common extrapolations (green lines), explicit regional climate models (circles), and adapted extrapolation (brown and blue line). Wind speed change between FOREST and GRASS is computed as an average over all onshore locations and 30 years. It is expressed relative to the change observed at the lowermost model level (ca. 30m). The upward sloping green lines with changes greater than 100% mean that the wind speed drop grows relative to the surface drop while the downward sloping lines indicate wind recovery with height. The common extrapolations use fixed values for the power and log law ($z_0 = 0.05m$, $\alpha = 0.143$), mimicking current common practise. By contrast, the adapted extrapolations use different roughness lengths in FOREST and GRASS (see Sec. 2.4) for details.





## 3.2 Wind change profiles per season at individual locations

Fig. 4 shows wind speed changes per height for different seasons and locations. We choose locations based on the amplitude of surface wind speed change, focusing on 90th, 50th (median) and 10th percentiles. That is, we analyze three locations where
winds are changed strongly, normally, and weakly by afforestation/deforestation for each model.

Both models agree that winter changes are generally highest, and summer changes are lowest, even though some exceptions exist. Moreover, the models also agree that wind speed changes decay monotonically with height although exceptions exists in summer, where IDL projects a local minimum followed by increased change (Fig. 4b,c).

While agreement on the amplitudes of wind speed changes 30m above ground is high in the 50th percentile (Fig. 4b, e) and
in the 90th percentile in summer (Fig. 4a, d), the models disagree strongly on other aspects. For instance, winter changes in the 90th percentile are twice as strong in IDL than GERICS (Fig.4a, d), and summer changes in the 10th percentile in IDL drop to about zero at 300m while they only marginally reduce from the near-surface value in GERICS (Fig.4c, f). In other words, model uncertainty is high at individual locations.

Comparing seasons reveals a strong seasonal dependency in IDL and a weaker but noticeable one in GERICS. The strongly
impacted location features convergence from vastly different values near surface to a similar range further up 4. Surface changes are very high in winter (ca. 3m/s), high in the annual mean and spring (ca. 2 m/s), medium in fall (ca. 1.75 m/s), and comparably low in summer (ca. 1.2m/s). By contrast, wind speed change is approximately 0.8 m/s at the highest displayed level (around 750m) in all seasons.

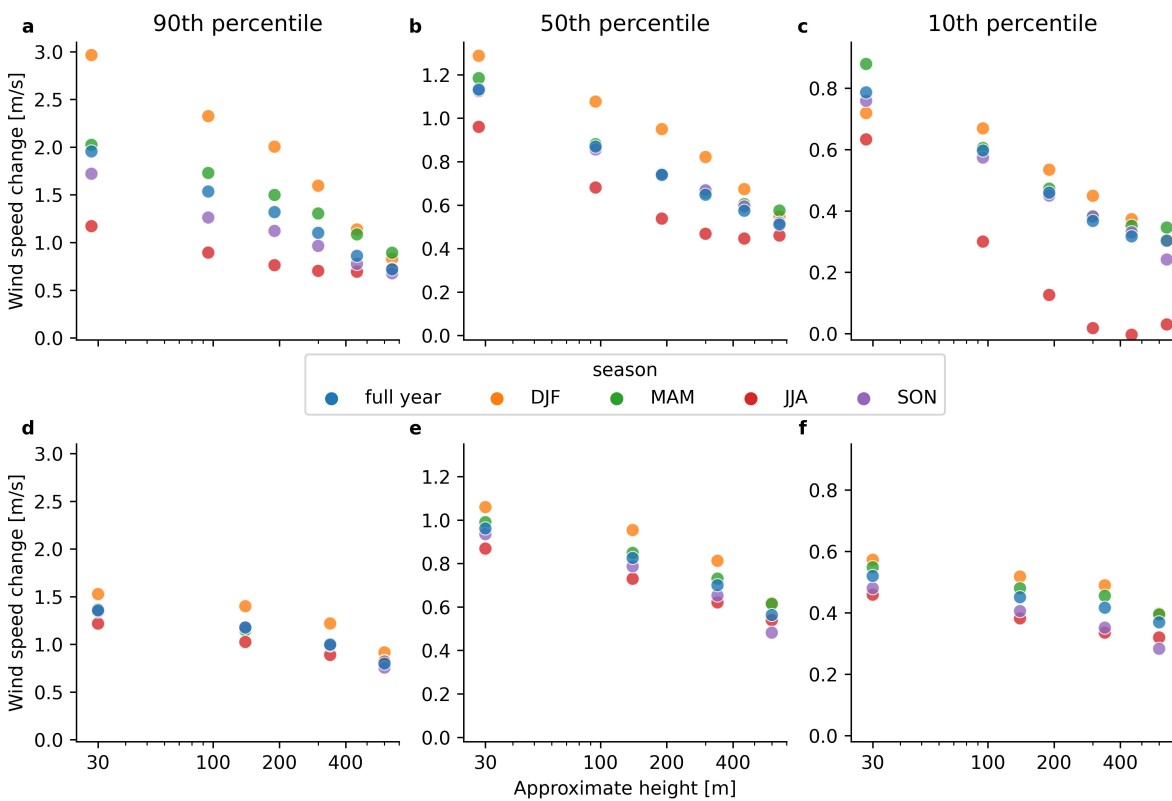

**Figure 4.** Wind profile changes between the GRASS and FOREST simulations over land according to IDL (upper row, a-c) and GERICS (lower row, d-f). Results are given for the full year, and individual seasons, respectively. The first (a, d) and last column (c, f) show wind speed change at a location that is strongly or weakly impacted by afforestation. The second column shows changes the median (50th percentile) location (b, e).



### 3.3 Case study: daily cycle of wind change profiles in different months

Besides the seasonal cycle, we expect changes in the daily cycle because sub-daily alterations in atmospheric stability and boundary layer height modulate the importance of the surface for upper-level winds. As an example, we analyze the daily cycle near Bielefeld in Central Germany (see Fig. S11 for a map) in different months, and report that the change signal indeed depends on the time of day and season (see Fig. 5).

In particular, we find that the surface perturbation decays slowly around noon and decays relatively quickly at night in all
months in IDL (5d,h,l,p) and in all months but October in GERICS (5b,f,j,n). While models roughly agree on time of day dependence, they disagree somewhat on the seasonal evolution. For instance, April changes are strong for GERICS but weak for IDL.

Moreover, changes in the daily cycle are more varied in IDL than in GERICS. For example, GERICS April changes are indistinguishable at 0h, 6h, 18h (Fig. 5f) while there are clear differences at those hours in IDL (Fig. 5h). Similarly, October
GERICS changes are identical at 0h, 6h, 12h while IDL shows quick decay with height at 0h, less decay at 6h, and almost no decay at 12h.

In addition, IDL features a local wind speed maximum at around 190m to 300m at midnight in April, July and October in the GRASS simulations (Fig. 5g,k,o) that does not exist in GERICS (Fig. 5e,i,m). This local maximum even gains additional strength in July from afforestation: the wind speed difference is between -0.4 m/s and -0.8 m/s (Fig. 5l). In other words, IDL
projects that wind speeds increase in the FOREST as compared the GRASS simulations at midnight in July. We will explore this unexpected feature, and evaluate whether it is a singularity at this location, in section 3.4.

Comparing the climatologies, we find that higher altitude winds generally blow stronger in GERICS than IDL (e.g., Fig. 5a vs. c). This is consistent with Breil et al. (2020) who found that IDL is warmer than GERICS, and is therefore expected to feature more mixing and reduce wind speeds further aloft.

Overall, we conclude that changes in the wind profile caused by afforestation/deforestation are highly complex. Constant extrapolation from the surface values would miss essential features of changes in the daily cycle, and is therefore not well suited to compute sub-daily hub height wind speeds needed for wind power conversion.



**Figure 5.** Wind speed profiles per hour of the day and month averaged over 2x2 grid boxes near to Bielefeld, Germany. The first (second) and third (forth) column display the wind speed (change) at model grid levels for GERICS and IDL, respecively. For example, the yellow box in **a** means that winds in January at 0 UTC and 600m above ground are between 13 and 14m/s in the GERICS GRASS simulations. Absolute wind speeds are shown in the color coding displayed on the lower left, while changes in wind speeds follow the color bar shown on the lower right. IDL data, which has hourly resolution, is only sampled every 6 hours to match the temporal resolution of GERICS.





### 3.4 Formation and extent of the summer nocturnal low-level jet in the IDL response

The higher wind speeds in FOREST as compared to GRASS are a large-scale phenomenon and not an artifact of the selected

case study location (see Fig. 6). While the wind speed difference is always positive at the lowest level at 28m, a band of negative values begins to emerge at 95m, strengthens up to 450m and decays thereafter. The band covers large parts of central Europe, including France, Benelux, Germany, Czech republic and Poland. It reaches values of up to -1.75 m/s which is considerable given that it occurs in summer when mean wind speeds are low (around 7m/s at 95m and 8m/s at 190m, see Fig S8d,g).

The vertical extent of this anomaly and its occurrence at nighttime in summer are in line with observed nocturnal low-

level jets (e.g., Weide Luiz and Fiedler, 2022) that occur as a consequence of decoupling of winds from the surface layer due to a temperature inversion. This nocturnal jet sits at a height that is relevant for wind energy. And it would be missed using the simple log or power law scaling, calling for explicit dynamical atmospheric modeling.

We can only speculate why the jet only occurs in IDL and not in GERICS. In addition to differences in the boundary layer schemes, one potential explanation is related to the temporal resolution that controls which processes can be explicitly

modeled. IDL has a model time step of 90s in the atmospheric model and the GERICS time step is 240s, which could mean that important processes are not resolved in the coarser GERICS simulations.

Moreover, the difference could also be related to the output frequency because nocturnal low-level jets are rather short lived, usually existing for a few hours. We can not tell from the available output whether GERICS captured the jet with a small temporal offset of a few hours. For example, if a jet occurred in the GERICS simulation at 2AM and lasted until 5 AM, we

would not be able to observe it from the 0AM and 6AM timesteps.

A process-based possible explanation is related to atmospheric stability. Davin et al. (2020) reported that summer temperature and surface fluxes differ more strongly in IDL than in GERICS between FOREST and GRASS. Moreover, Breil et al. (2020) evaluated the July temperature change in France in the same model simulations, finding that nighttime FOREST temperatures increase slightly compared to GRASS in the lowest atmospheric model and at the surface in REMO while they decrease

in IDL (see their Fig. 8g, where WRF-NoahMP is almost identical to the setup used by IDL). The temperature drop in the lowest atmospheric level could be indicative of greater atmospheric stability, although a full analysis of the vertical column would be neccessary to confirm this hypothesis, which is beyond the scope of the current study. The higher IDL summer temperatures over FOREST could imply that the boundary layer is more stable than in GERICS, favouring the creation of nocturnal low-level jets.

While we can not robustly pinpoint a single explanation for the jet occurrence, our results emphasize the need for explicit modeling rather than extrapolation of surface winds, as low-level jets would generally not be captured using the log and power laws.





**Figure 6.** Wind speed change at the IDL model levels evaluated at midnight (0 UTC) in July. Change is computed as GRASS - FOREST. Red values correspond to higher wind speeds in GRASS compared to FOREST, and blue corresponds to higher winds in FOREST compared to GRASS. Subplot titles give the approximate height per model level, increasing from **a** to **f**.



## 3.5 Afforestation impacts on wind power generation

Figure 7 provides maps of mean capacity factors (CFs) in both experiments as well as their difference. Both models feature a
clear land-sea divide in both scenarios yet the absolute CFs differ between the models. While GERICS has a large area of very
high offshore CFs exceeding 0.65 North West of the UK, IDL shows largest CF further to the East and IDL CFs almost never
exceed 0.65 (see Fig. 7a,d). Over land, the models swap roles and GERICS CFs are generally lower than those from IDL.

We tested whether this inter-model difference is caused by the different output frequencies in combination with the non-
linear power curve. Power generation could be underestimated in calm onshore locations and overestimated in windy offshore
locations when using 6 hourly output because the tails of the distribution are less well sampled compared to hourly output.
However, we found that the IDL mean capacity factors are not substantially impacted by downsampling to 6 hourly resolution
(Fig. S15) and thus conclude that the output frequency difference does not explain the strong inter-model differences.

Irrespective of those differences, both models agree that afforestation substantially reduces CFs by more than 0.05 in most
locations (Fig. 7c,e and 8). CF reductions even exceed 0.11 over approximate half of the continent in GERICS. That is, a single
3.6 MW turbine would produce about 3.5 GWh/y less in the FOREST simulation. The reduction is particularly striking when
considered relatively: the GRASS CF is more than 45% higher than the FOREST CF (Fig. S14). The reductions are weaker in
most locations in the IDL simulation, and tend to be in the range between 0.05 to 0.1. Exceptions are Norway, Southern Spain,
and Turkey where IDL projects stronger reduction potentially related to the different suitability criteria for tree growth. The
combination of higher mean CFs and lower changes means that the relative reduction is markedly lower in IDL and CFs often
drop by around 10% in Northern Europe. However, drops still reach values of 30% in Southern Europe, thus being clearly
non-negligible.

The distribution of mean CFs is wider for IDL than for GERICS (Fig. 8). For example, in the GERICS FOREST simulation,
the majority of grid cells feature CFs between 0.2 and 0.3, and an individual 0.01 CF increment occurs in more than 10%
of the grid cells. By contrast, the IDL FOREST simulations cover values from 0.1 to 0.6 more evenly and no single 0.01 CF
increment occurs in more than 5% of grid cells. Moreover, while GERICS features a single distinct peak in both experiments,
IDL is weakly bimodal.

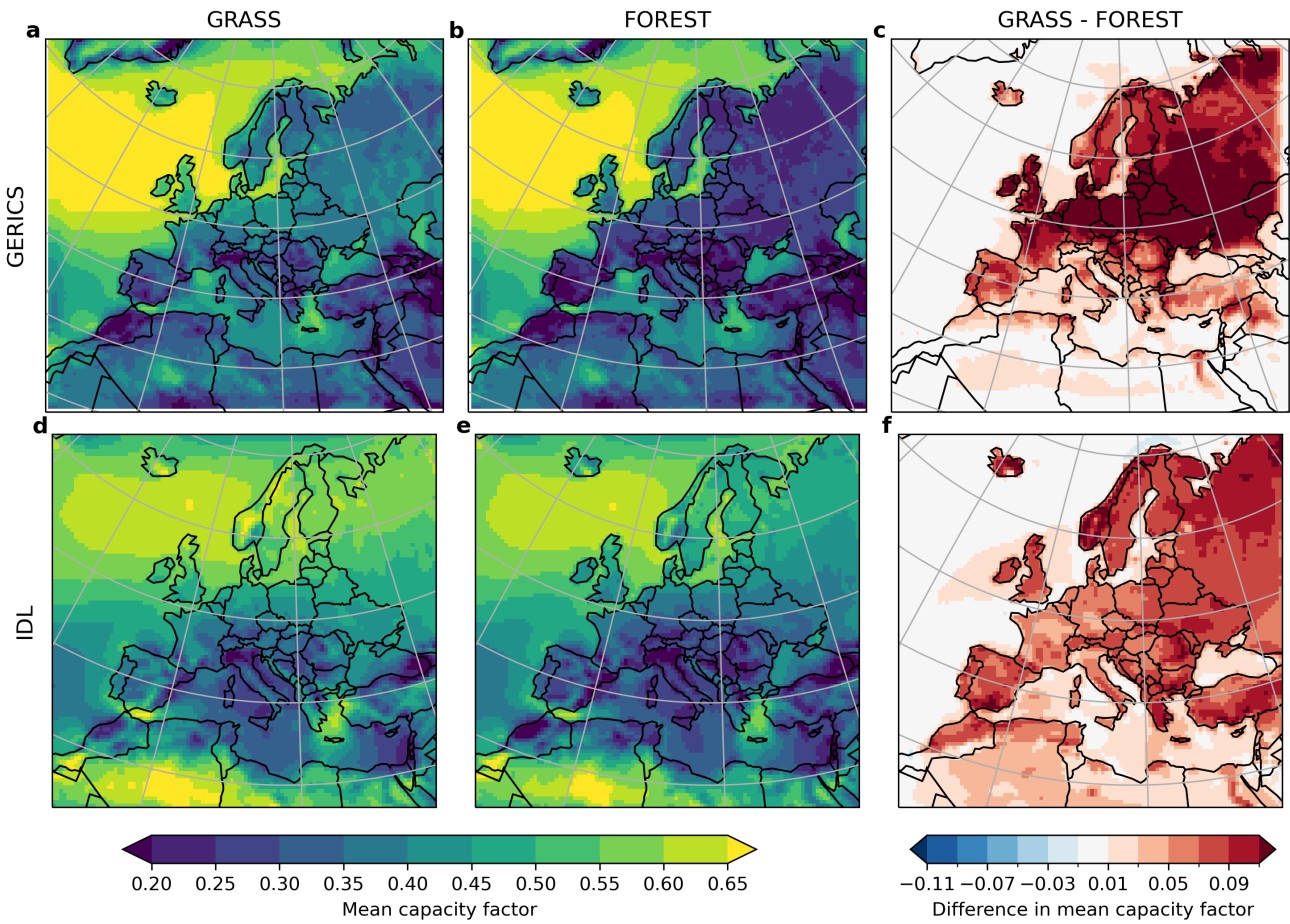

**Figure 7.** Mean capacity factors for the SWT120-3600 turbine in the GRASS (**a,d**) and FOREST (**b,e**) simulations, as well as the GRASS-FOREST difference (**c,f**) in capacity factors. Note that the domain sizes are slightly different because the GERICS simulation provide a few additional grid boxes outside the core domain.




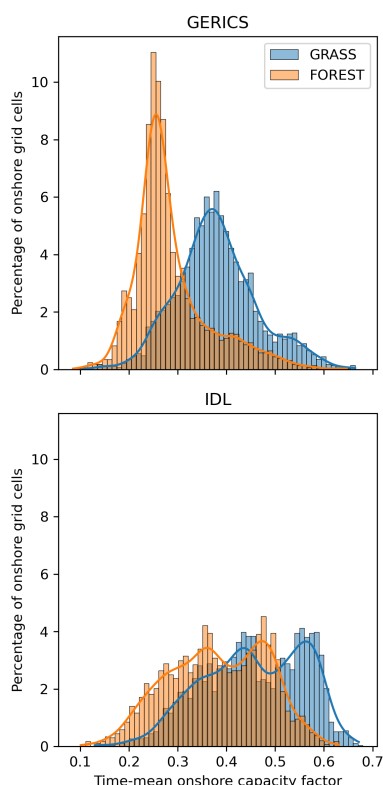

**Figure 8.** Distribution of European onshore capacity factors by model (**a** vs. **b**) and experiment (blue vs. orange). Values are normalized such that the sum over all bins equals 1.



## 4    Conclusions

Using two regional climate model simulations from the CORDEX Flagship Pilot Study LUCAS (Davin et al., 2020), we have shown that afforestation substantially impacts winds and wind energy capacity factors. Changes remain relevant everywhere in the lowermost 350m, even exceeding 1 m/s at around 300m, and should therefore not be ignored in wind energy assessments.

Different parts of the day and year are affected differently, with implications for power system design. While afforestation impacts higher level winds only weakly over night, it impacts them more strongly around noon, when current electricity demand typically peaks. Similarly, winter winds are more strongly affected than summer winds, implying that the benefits of using wind power to compensate generation shortfalls from solar photovoltaics in winter might be compromised. Using a 3.6MW wind turbine with 120m hub height, we find that capacity factors are up to 50% higher in the GRASS as compared to the FOREST scenario according to one model (GERICS) and up to 30% according to the other model (IDL).

The reported changes are complex, non-local, depend on the location, impact the daily and seasonal cycle, and are therefore not well captured by simple heuristics like the log law and power law even when using modified parameters. Using the log and power law in simulations with substantially different land use with the same parameters is particularly detrimental because it overestimates the effects of land use change on hub height winds, and might mask dynamical changes. We therefore strongly recommend to not scale 10m climate model output to hub heights with constant coefficients in the log or power law.

Instead, model level information could be used as done in Hahmann et al. (2022) or in the present study. However, using model level outputs is generally more challenging because it requires in-depth knowledge about the structure of climate models (e.g., terrain-following vs. absolute vertical coordinates) and is more data-heavy for impact modelers. The provision of wind speeds interpolated to multiple heights in the 100m to 200m band by climate modeling groups would therefore enable better climate (change) assessments for wind energy and help to overcome the disconnect between energy and climate modeling (Craig et al., 2022).

While covering Europe completely in forest or grass are clearly edge cases that are unlikely to happen in reality, strong land use changes do exist in the widely used simulations from the Climate Model Intercomparison Project (Taylor et al., 2012; Eyring et al., 2016), for example, over Eastern Europe in the representative concentration pathway 4.5 (e.g., Wohland, 2022). The results of this study thus have implications for CC impact assessments based on CMIP or CORDEX.

The results in this study are restricted to two models that provided data at the required granularity in the vertical and time dimensions. Since we also report substantial model uncertainty, the exact values provided here should be treated with caution, even though the overall results are consistent and physically plausible. Future work, however, should focus on including more models and scenarios and we intend to complement this study with different models and more realistic scenarios from the second phase of the LUCAS experiment.



*Code and data availability.* The code is written in python and maintained on github. It will be made openly available upon publication.

*Author contributions.* JW developed the research idea, implemented it, produced all figures, wrote the code, and drafted the manuscript. PH and DR gave feedback about the REMO model. OA and DL provided data. MB helped to contextualize the results. All authors reviewed and
edited the manuscript, and attented progress discussions.

*Competing interests.* We declare no competing interests

*Acknowledgements.* We would like to thank the wider LUCAS consortium for discussion, in particular Merja Tölle, Edouard Davin, Priscilla Mooney, Pedro Soares, and Eleni Katragkou. The authors gratefully acknowledge the WCRP CORDEX Flagship pilot study LUCAS (Land use and Climate Across Scales), and the research data exchange infrastructure and services provided by the Jülich Supercomputing Centre,
Germany, as part of the Helmholtz Data Federation initiative. Jan Wohland is part of the ETH Zurich SPEED2ZERO initiative which received support from the ETH-Board under the Joint Initiatives scheme. Daniela C.A. Lima was supported by the Scientific Employment Stimulus 5th edition from Fundação para a Ciência e a Tecnologia (FCT, 2022.03183.CEECIND).



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
