# Peer review of "Extrapolation is not enough: Impacts of extreme land-use change on wind profiles and wind energy according to regional climate models"

_EGUsphere, 2023_

## Author Response (AR1)

Response to the reviewers

**Extrapolation is not enough: Impacts of extreme land-use change on wind profiles and wind energy according to regional climate models**

Jan Wohland, Peter Hoffmann, Daniela C. A. Lima, Marcus Breil, Olivier Asselin, and Diana Rechid

Under review for ESD

We would like to thank both reviewers for their thoughtful, critical and clear assessment of our submission. We addressed all comments and believe that the quality of the manuscript has improved during the process. We are confident that we addressed all reviewer comments convincingly and provide detailed responses to the individual comments below.

Throughout this document, we use *italics* to mark the reviewer comments,  to mark text that we deleted from the manuscript and blue to mark text additions to the manuscript. When reporting line numbers, we refer to the initial submission.

**Reviewer 1**

*I have reviewed the paper titled "Extrapolation is not enough: Impacts of extreme land-use change on wind profiles and wind energy according to regional climate models". I believe the paper needs major revision as more details and analysis are needed to justify what the authors mentioned. In addition, the language needs sharpen to facilitate better understanding for the reader.*

**Author response**

We would like to thank the reviewer for their skeptical assessment of our manuscript. We address the individual comments – as well as changes to the manuscript inspired by them one-by-one below.

**Comment 1-1**

*Lines 125-126: remove "uses modeling time steps of 90 seconds, while GERICS uses 240 seconds". This is one of the many incidents where the sentence is awkward. The language needs more polish.*

**Author response**

Thanks for the comment. We agree that the sentence can be removed here as the information is repeated in the Results & Discussion section (l. 270) as a potential reason for the inter-model spread.

Following your comment, we also checked the entire manuscript again (also by a native speaker) and made editorial changes to improve the language. We refer to the track changes version of the manuscript where these changes are documented.

**Changes to the manuscript**

l. 126

> … GERICS.

**Comment 1-2**

*More details are needed in Table 1(or describe in text) to show the readers the difference between the two models, such as the boundary layer scheme, surface layer scheme, land surface model and the boundary forcing.*

**Author response**

Thank you for this comment. The information that you asked for is provided in the referenced publication that introduces the LUCAS dataset (Davin et al., 2020), namely in their Table 1. We had initially decided to keep this level of detail out of our own manuscript to increase legibility, avoid duplication and because the dataset is not developed as part of this publication. However, we agree with you that more detail about the differences between the models will be useful for some readers to clarify the experimental setup and we therefore decided to add a pointer to the Table for interested readers (see below).

**Changes to the manuscript**

l. 96:

> … the institution (i.e., GERICS or IDL). Specifics of the model setups like the lateral boundary formulation or the boundary layer scheme are documented in Table 1 in Davin et al. (2020). We acknowledge …

**Comment 1-3**

*Figures 1 and 2: can you provide an explanation why the difference in wind speed reduces from lower to higher vertical level?*

**Author response**

Yes. This behavior is expected. The trees in the FOREST simulation reduce horizontal momentum near the surface via increased friction. This effect is strongest close to the surface, and it becomes irrelevant in the free troposphere. The results in Figs. 1 & 2 show the transition between those two situations and highlight that the effect of forests remains sizeable even about 300m above ground. We add a sentence to explain our expectation, as other readers might also wonder about the same thing.

**Changes to the manuscript**

l. 163:

> In line with physical expectation, the effect of forests is strongest near the surface where momentum is removed from the atmospheric flow and weakens with height, ultimately becoming irrelevant in the free troposphere. Our core finding is that afforestation/deforestation matters for winds around hub height according to both models.

**Comment 1-4**

*Line 167-168* **"IDL features reductions of similar magnitude, even exceeding 1.5 m/s in large parts of Scandinavia and Iceland near the surface that slowly decay with height (Fig. 2l)."** *Do you mean Figure 2?*

**Author response**

We mean Figure 2 panel l. Thanks for pointing out that this could also be misread as, for example, Figure 21 (i.e., twenty-one). We changed this instance and all others by adding a ", panel" numbers and letters in figures references and we expect that the final typesetting by the journal will further improve this technical aspect.

**Changes to the manuscript**

In all instances, we add a space in references to Figumailre panels:

> Fig.  N, panels j

**Comment 1-5**

*Lins 178-180* **"This uncertainty is a combination of at least two factors, namely (a) how the models (or modelers) translate the scenario into actual boundary conditions and (b) how the models respond to those boundary conditions."** *Given the differences in model configuration and run time option, this statement is not accurate. I suggest remove this sentence.*

**Author response**

Thanks for the comment. It is true that this sentence may not emphasize enough that other sources of uncertainty are possible. To avoid further misunderstanding, we rephrased it to emphasize that we focus our attention on two of many possible factors driving model uncertainty.

**Changes to the manuscript**

>  Among the numerous factors influencing model uncertainty, two are of particular relevance here, namely (a)

**Comment 1-6**

*Section 3.2/Figure 4: why just 3 locations? The result is statistically insignificant. More points are needed for the analysis. For instance, the authors should randomly select at least 10 points per percentile range to get an estimate.*

*Figure 5/ Lines 255-258: Again, more data points are need for the analysis.*

**Author response**

Thanks for those two comments, which we cluster together as they address related issues.

We agree that the results are not statistically significant, and we would like to stress that we made this very clear in the writing. Even the section headings stressed the exemplary nature of this part of the analysis:

"3.2 Wind change profiles per season at **individual locations**"

"3.3 **Case study**: daily cycle of wind change profiles in different months"

Despite the lacking significance tests, these results provide evidence that wind profile (changes) are complex. Standard extrapolation is incapable to capture these features. For instance, it would scale near-surface wind speed reductions in the same way throughout the day (in stark disarray with the case study presented in Fig. 5) and across seasons (in disarray with the individual locations presented in Fig. 4). The presented evidence therefore backs up our conclusions.

Nevertheless, we implemented your suggested methodology and provide an updated version of Figure 4 below. Please note that the results are virtually identical, and that the sampling uncertainty is very small to negligible in all cases. To be specific, we randomly chose 10 locations that lie within the 90±5; 50±5, 10±5 th percentile and compute the mean and standard deviation of wind speed change per height and model. These relatively wide percentile ranges ensure that the risk of sampling the same location multiple times is very low as bands encompass about 290 locations.

We then plot the means and add the standard error of the mean as errorbars (i.e., standard deviation / sqrt(N)), see next page. Comparing the old and new version of the Figure, we conclude that changes in the mean are minor and do not impact our conclusions. Moreover, the error bars are generally very small. It is even impossible to identify them in the plot for many data points. Even in those cases where the error bars are relatively large, for example, at the 90th percentile at 30m in DJF or at the 10th percentile around 400m in JJA, they remain irrelevant when compared to inter-season spacing or the evolution with height. We add a brief discussion of this additional statistical analysis - as well as an explanation of the method - to the manuscript, see below.

We decided to not add further analysis to the case study presented in Fig. 5 because it is clearly flagged as a case study and a case study is sufficient to justify the drawn conclusions. Moreover, we would like to point out that we already included two additional locations in the Supplementary Material. Thanks to your comment, we realize that we haven't properly referenced them in the manuscript. We therefore fix this shortcoming by adding a brief discussion as part of the revisions (see below).

**New Fig. 4**

[Figure]

[Figure]

**Old Fig. 4**

**Changes to the manuscript**

Caption of Fig. 4

Wind profile changes between the GRASS and FOREST simulations over land according to IDL (upper row, a-c) and GERICS (lower row, d-f). Results are given for the full year, and individual seasons, respectively. The first (a, d) and last column (c, f) show wind speed change at a location that is strongly or weakly impacted by afforestation. The second column shows changes the median (50th percentile) location (b, e). Points denote the mean over 10 randomly sampled locations in a centered percentile band with a width of 10 percentiles, and error bars denote the standard error of the mean.

l. 217:

**3.2 Wind change profiles per season at strongly, normally, and weakly impacted  locations**

Fig. 4 shows wind speed changes per height for different seasons and locations. We choose locations based on the amplitude of surface wind speed change, focusing on 90th, 50th (median) and 10th percentiles. That is, we analyze three locations where winds are changed strongly, normally, and weakly by afforestation/deforestation for each model.

To ensure that our results are representative for a range of similar locations rather than artifacts of a single location, we draw random ten-member samples from a centered percentile range with a width of ten percentiles. For instance, the $90^{th}$ percentile corresponds to a 10-member draw from the percentile range between the $85^{th}$ and $95^{th}$ percentile, and the same approach is used for the other cases (i.e., $50^{th}$ percentile: $45^{th}$-$55^{th}$; $10^{th}$: $5^{th}$-$15^{th}$). We report the mean change per season and model together with error bars representing the standard error of the mean (i.e., std/sqrt(N), where std is the standard deviation of wind speed change across the 10 locations from each percentile band and N=10 is the sample size).

Both models agree that winter changes are  highest, and summer changes are lowest, even though some exceptions exist. Moreover, the models also agree that wind speed changes decay monotonically with height although exceptions exists in summer, where IDL projects a local minimum followed by increased change (Fig. 4b,c). Error bars are small compared to the difference between seasons, vertical levels, and models, indicating the robustness of the presented results to sampling different yet similar locations. In many cases, the error bars do not even extend beyond the marker for the mean value.

p. 226

… and summer changes in the 10th percentile in IDL drop below zero at 300m

p. 229

The strongly impacted location features convergence from vastly different values near surface to a similar range further up (Fig. 4a).

p. 255

… further aloft.

We analysed additional locations in Spain and Sweden and report that those also feature complex responses of the wind profile (see SI Figs. D2 and D3), strengthening the conclusions drawn from the German case study. While some effects co-occur at all three locations, such as particularly strong upper level changes in IDL at noon, other effects are unique for the respective sites, suggesting that explicit modeling of the vertical wind field structure is needed.

Overall, we conclude that changes in the wind profile caused by afforestation/deforestation are highly complex.

SI Figure Caption D2
Same as Fig. 5 but for 2x2 grid boxes in western Spain (near Madrid, see Fig. D1 for a map).

SI Figure Caption D3
Same as Fig. 5 but for 2x2 grid boxes in south-western Sweden (near Gothenburg, see Fig. D1 for a map).

**Comment 1-7**

*Lines 259 to 260:* **"The higher wind speeds in FOREST as compared to GRASS are a large-scale phenomenon and not an artifact of the selected case study location (see Fig. 6)"** *Why is it reasonable for the wind speed to be higher in FOREST than GRASS? What larger-scale phenomenon are you referring to?*

**Author response**

Thanks for this question which we asked ourselves as well. Why would winds be stronger over a forested area than over a grass-covered area? The answer has two components. First, please note that Fig. 6 shows midnight July winds and that average winds over all months and all hours of the day are weaker over forests than over grass (cf. Fig. 2). That is, the forest reduces wind speeds on average. But the forest also strengthens winds during specific parts of the year.

Second, the mechanism that can explain higher winds over forests is already mentioned in the text:

"The vertical extent of this anomaly and its occurrence at nighttime in summer are in line with observed nocturnal low-level jets (e.g., Weide Luiz and Fiedler, 2022) that occur as a consequence of decoupling of winds from the surface layer due to a temperature inversion."

Basically, the theory says that less horizontal momentum is transported downwards during the temperature inversion. Consequently, winds at 95m to 650m can be higher despite higher momentum removal near the surface, as demonstrated by the model output.

To clarify the language, we use the term "large-scale phenomenon" here in contrast to the case study analysis presented in Sections 3.2 and 3.3 which – as you point out – are just individual examples. As apparent in Fig. 6, however, wind speeds are higher in FOREST than in GRASS in a large area covering multiple countries in Central Europe. In other words, this jet-like response is not an artifact of the sampling in Sect. 3.2 & 3.3. Instead, it is characteristic for a relevant portion of the studied domain.

**Comment 1-8**

*Line 284-285:* ***"The higher IDL summer temperatures over FOREST could imply that the boundary layer is more stable than in GERICS, favouring the creation on nocturnal low-level jets"*** *Why higher temperature favors stable boundary layer?*

Thanks for this question. We realize that our explanation was not clear enough. The reasoning is as follows.

During the day, the surface is heated by the sun and releases a part of this energy into the atmosphere in the form of sensible heat fluxes, thereby warming the boundary layer. In IDL, these sensible heat fluxes are significantly higher than in GERICS (Davin et al., 2020), which results in a stronger heating of the boundary layer during the day.

During the night, temperatures generally decrease stronger at the surface than in the boundary layer, due to the outgoing longwave radiation at the surface. Consequently, the surface cools the boundary layer from below, and temperatures increase with height and thus a stable stratification evolves near the ground. As the boundary layer in IDL has warmed up stronger during the day, the nocturnal temperature gradient between the cooling surface and the atmosphere is greater than in GERICS and the atmospheric stratification is therefore more stable, favoring the development of a nocturnal low-level jet.

**Changes to the manuscript**

l. 276f

> We propose a potential  process-based possible explanation  related to atmospheric stability. During the day, the sun heats the surface which releases some of the heat into the atmosphere as sensible heat fluxes, thereby warming the boundary layer. These fluxes are significantly higher in IDL than in GERICS (Davin et al., 2020), resulting in a stronger daytime boundary layer heating. During the night, temperatures generally decrease stronger at the surface than in the boundary layer, due to the outgoing longwave radiation at the surface. The surface therefore cools the boundary layer from below, and temperatures increase with height and thus a stable stratification evolves near the ground. As the boundary layer in IDL has warmed up stronger during the day, the nocturnal temperature gradient between the cooling surface and the atmosphere is greater than in GERICS and the atmospheric stratification is therefore more stable, favoring the development of a nocturnal low-level jet. While this explanation is physically plausible and in line with previous results (e.g., Breil et al., 2020),

~~Moreover, Breil et al. (2020) evaluated the July temperature change in France in the same model simulations, finding that nighttime FOREST temperatures increase slightly compared to GRASS in the lowest atmospheric model and at the surface in REMO while they decrease in IDL (see their Fig. 8g, where WRF-NoahMP is almost identical to the setup used by IDL). The temperature drop in the lowest atmospheric level could be indicative of greater atmospheric stability, althoughThe higher IDL summer temperatures over FOREST could imply that the boundary layer is more stable than in GERICS, favouring the creation of nocturnal low-level jets.~~

While we can not robustly pinpoint a single explanation…

**Reviewer 2**

*I appreciate the chance to review Extrapolation is not enough: Impacts of extreme land-use change on wind profiles nad wind energy according to regional climate models. Furthermore, I appreciate the exploration of from such extreme model scenarios as 'all grass' or 'all forest' onshore and how it affects land-atmosphere interactions and vertical wind profiles. However, the method is not reproducible as written requiring me to look elsewhere for answers and then I found a major error, and I also have some reactions to how the reason for the study and the associated results are presented.*

**Author response**

Thanks for the assessment and the valuable feedback. We are glad that you spotted the mistake related to the turbine hub height which we fixed now. Our main results remain unchanged. We address your individual concerns, and how we solved them, below.

We are surprised that you consider our method to not be reproducible. We believe to have explained all necessary parts of the analysis in the manuscript. We are happy to share our code pre-publication with you if you consider this useful to evaluate our work.

*Reviewer comment 2-1*

***The major error relates to the use of the SWT120-3600 turbine specifications.*** *According to the manufacturer's website (https://en.wind-turbine-models.com/turbines/669-siemens-swt-3.6-120-offshore the hub-height of this wind turbine is either 88 or 90 meters, with a 90-meter hub-height also stated here (https://www.thewindpower.net/turbine_en_79_siemens_swt-3.6-120.php) and here (https://github.com/wind-python/windpowerlib/blob/dev/windpowerlib/oedb/turbine_data.csv). The manuscript states (line 160-1) "This turbine has a hub-height of 120 meters and was chosen following Wohland et al. (2021a) as it represents the median current wind turbine." In the three web references in this paragraph, the '120' refers to the rotor diameter not the hub-height. It remains possible the author is using a difference reference for the hub-height, or it could simply be an honest mistake. Assuming it is in fact an error, figures (Fig. 7&8) and sections of the text related to the capacity factor are wrong and require complete revision.*

**Author response**

Thanks for spotting this mistake. We corrected the hub height from 120m to 90m and re-ran the full analysis which left the main conclusions qualitatively unchanged. As a consequence of the hub height change, IDL is now interpolate between the two lowermost model levels (ca. 28m and 95m) while the wrong hub height was interpolated from one level higher up (ca. 95m and 190m). We provide the new Figures below and discuss the changes per Figure. Impacted figures are Fig. 7 & Fig. 8 in the manuscript as well as Supplementary Figs. A1, E1 & F1. The following pages are structured as:

1. New Fig.

2. Old Fig.
3. Figure caption copied from the manuscript
4. Comment about the impact on our analysis
5. Page break

Lastly, we provide changes to the manuscript jointly for all Figs.

As a side note, we would like to point out that the turbine was chosen because it is median in terms of the sensitivity of its capacity factors to wind speed changes. Specifically, we evaluated the windpowerlib turbines at 7 m/s and chose the turbine that has the median generation at that wind speed. In other words, the choice of the turbine is not affected by the lookup mistake of the turbine hub height.

**New Fig. 7: Mean capacity factors for the SWT120-3600 turbine**

[Figure]

**Old Fig. 7 (from initial submission)**

[Figure]

**Figure caption**

Mean capacity factors for the SWT120-3600 turbine in the GRASS (a,d) and FOREST (b,e) simulations, as well as the GRASS-FOREST difference (c,f) in capacity factors. Note that the domain sizes are slightly different because the GERICS simulation provide a few additional grid boxes outside the core domain.

**Impact on our analysis**

Mean capacity factors are reduced in the new Fig. 7 as compared with the old one. This reduction is expected because the corrected hub height is 30m lower than the initially used one, implying that hub height winds were initially overestimated. The effect shows very strongly in the GERICS FOREST maps in Central Europe (compare Benelux, Germany, Poland in Fig. 7b new and old) and also shows in other parts of the analysis, such as the GRASS-FOREST difference plots in Southern France using the IDL simulations (Fig. 7f). We refer to the next Figure for a more quantitative analysis.

As far as the big picture is concerned, however, the main message remains unchanged: land-use change exerts a strong control over wind power potentials according to these regional climate models.

**New Fig. 8: Distributions**                    **Old Fig. 8 (from initial submission)**

[Figure]

[Figure]

**Figure caption**

Distribution of European onshore capacity factors by model (a vs. b) and experiment (blue vs. orange). Values are normalized such that the sum over all bins equals 1.

**Impact on our analysis**

While maintaining their key characteristics, the distributions are shifter to lower mean capacity values, in line with physical expectation for a reduced hub height (please note the slightly modified x-axis range).

In particular, the single maximum in the GERICS simulations and the bi-modal shape in the IDL simulations remains unchanged and the shift between the GRASS and FOREST experiments remains similar.

We conclude that both Figures from the main manuscript are only impacted moderately by the hub height correction. Nevertheless, we modify the presentation of results as documented at the end of this comment.

**New Fig. S1: Example of wind speed interpolation to hub heigh**

The only change to the old Fig. S1 is that the position of the stars (i.e., hub height) has been shifted from 120m to 90m. We therefore restrain from showing the old Fig here as well.

**Figure caption**

Example of wind speed interpolation to hub height. Wind speeds at 90m (stars) are computed from the closest two model levels (circles) by fitting the power law exponent α at each time step and each location. The subplot on the left shows 8 example profiles. The subplot on the right shows the distribution of power law exponents during one example timestep.

**Impact on our analysis**

This plot illustrates that 90m sits fairly centered between the lowermost two model levels (here for GERICS), implying that the method remains equally suitable as when we used it for 120m winds. For IDL, we now use the 2 lowermost model levels to interpolate from about 30m and 95m to 90m.

This illustrative plot documents why the change from 120m to 90m hub height only has relatively small impact on the essence of the results. While the correction of course matters, it introduces a relatively small change in hub height winds only because the wind profile is reasonably flat in the 90m to 120m domain, as shown in the plot above on the left-hand side.

**Relative changes of mean capacity factors**

**New Fig. S14**

**Old Fig. S14**

[Figure]

[Figure]

**Figure caption**

Relative changes of mean capacity factors for GERICS (a), IDL (b), and IDL resampled to 6h values (c).

**Impact on our analysis**

The updated Fig. S14 illustrates that the difference between the GRASS and FOREST simulations is stronger when using a 90m hub height (New Fig. S14) as compared to 120m (Old Fig. S14), as expected. Nevertheless, key take-home messages remain unaffected from this change. First, the GERICS simulations feature stronger capacity factor changes between the forested and grass-covered scenarios particularly in the Northern half of the continent. Second, resampling IDL to 6h has negligible effects on the mean capacity factors (compare Fig. S14 b & c).

**New Fig. S15**

[Figure]

**Old Fig. S15**

[Figure]

**Figure caption**

Same as Fig. 7 but the IDL data (d-f) has been downsampled to 6 hourly before taking the temporal mean.

**Impact on our analysis**

The change from the Old Fig. S15 to the new one introduces basically the same changes as the one from the Old Fig. 7 to the new Fig. 7 (see above).

**Changes to the manuscript (all new Figs combined)**

l. 160

This turbine has a hub height of 90 meters and was chosen following Wohland et al. (2021a) as it represents the median current wind turbine in terms of its capacity factor at 7m/s.

l. 300

The reduction is particularly striking when considered relatively: the GRASS CF is more than 50% higher than the FOREST CF in most European onshore locations (Fig. S14). The reductions are weaker in most locations in the IDL simulation, typically ranging between 0.07 and 0.11 . Exceptions are Norway, North-Western Russia, Southern Spain, and Turkey where IDL projects stronger reduction potentially related to the different suitability criteria for tree growth (Fig. 7). The combination of higher mean CFs and lower changes implies  that the relative reduction is markedly lower in IDL compared to GERICS, particularly in Northern Europe.  However, drops still reach values of 40% in Southern Europe in IDL, thus being clearly non-negligible.

The distribution of mean CFs is wider for IDL than for GERICS (Fig. 8). For example, in the GERICS FOREST simulation, the majority of grid cells feature CFs between  0.15 and 0.25, and an individual 0.01 CF increment occurs in more than 10% of the grid cells.

l. 319

Using a 3.6MW wind turbine with 90m hub height, we find that capacity factors are up to 50% higher in the GRASS as compared to the FOREST scenario according to one model (GERICS) and up to 40% according to the other model (IDL).

SI, page 1

**Fig. S1** Example of wind speed interpolation to hub height. Wind speeds at 90m (stars) are computed from…

**Reviewer comment 2-2 (we broke your bullet point list into individual comments to make it easier for us to refer to them)**

*Title: what do you suggest given that Extrapolation is not enough…?*

*It might be assumed from the focus in the manuscript that a model is appropriate, but then, given that you are showing how the swapping of all grass to all forest influences the hub-height wind speeds, what should be expected from large-scale installations of wind turbines over grass or forest; this isn't the exact point of your paper but it does undermine the applicability of these model simulations to your suggested application (as per the Abstract and Conclusion)*

**Author response**

Thanks for this comment and we are happy to clarify. Our main point is that climate-model based wind energy assessments should use model output at heights close to hub height. In this study, we use regional climate model output at native model levels and show that the changes in the wind profile depend on time and location and are thus not well captured by simplified extrapolations like the log and power law, which assume independence of location and time. Since using native model levels is technically demanding (see our response to your comment 2-4 as an example), for example, for someone with no prior knowledge about climate models, we recommend that climate modeling groups save winds at multiple relevant heights.

We summarized this conclusion in the manuscript (l. 325f) and copy the relevant sentences below:

"We therefore strongly recommend to not scale 10m climate model output to hub heights with constant coefficients in the log or power law.

Instead, model level information could be used as done in Hahmann et al. (2022) or in the present study. However, using model level outputs is generally more challenging because it requires in-depth knowledge about the structure of climate models (e.g., terrain-following vs. absolute vertical coordinates) and is more data-heavy for impact modelers. The provision of wind speeds interpolated to multiple heights in the 100m to 200m band by climate modeling groups would therefore enable better climate (change) assessments for wind energy and help to overcome the disconnect between energy and climate modelling (Craig et al., 2022)."

We hope that this explanation helped to clarify the issue.

**Reviewer comment 2-3**

*Abstract (line 11-13): "Our results confirm earlier studies that land use change impacts on wind energy can be severe and that they are generally misrepresented with common extrapolation techniques."*

*As the concluding sentence of the Abstract, what new knowledge will be presented to the reader here?*

*? Are the models analyzed here better or different than the European observations (Vautard et al. 2010 in Nature Geoscience): "In addition, mesoscale model simulations suggest that an increase in surface roughness—the magnitude of which is estimated from increases in biomass and land-use change in Eurasia—could explain between 25 and 60% of the stilling. Moreover, regions of pronounced stilling generally coincided with regions where **biomass has increased over the past 30 years, supporting the role of vegetation increases in wind slowdown."***

**Author response**

Thanks for this comment.

The Vautard et al. (2010) stilling paper and more recent ones, like the follow-up study by Zeng et al. (2019) draw from measurements taken at weather stations and are thus different than the study currently under review. Please also note that the initial attribution of stilling to increases in surface roughness is challenged by the more recent reversal of stilling (see Zeng et al., 2019) and the fact that stilling and reversal-of-stilling phases occur without any forcing in climate model simulations (Wohland et al., 2021).

The new knowledge provided in this study is that land-use change still matters at hub height. Please note that both Vautard et al (2010) and Zeng et al. (2019) base their assessment on 10m winds (a standard height for meteorological measurements) and do not provide analyses for upper level winds. The results of the present study are, however, most relevant for the assessment of future wind resources as future scenarios contain significant amounts of land-use change.

**Reviewer comment 2-4**

*Methods*

*Are you sure the model heights you use throughout are midpoints of the vertical rather than the top of each model height (Fig. 4&5) At least for WRF, these heights could be post-processed differently from how the model considers altitude.*

**Author response**

Thanks for this fantastic comment.

In the GERICS simulations, there is indeed an offset of half a grid box between geopotential and horizontal winds: horizontal winds are reported on numbered levels *lev* but geopotential is unknown at those heights. Geopotential is instead reported on shifted (half a vertical grid cell) *lev_2* levels. After consultation with the GERICS modelers, we decided to linearly interpolate the geopotential values from the *lev_2* to the *lev* coordinate.

We faced a similar issue in the horizontal dimension as well, where GERICS reports wind components *U* and *V* on a shifted grid. Specifically, *U* is reported at the western and eastern

edge of the grid cell and centered in the north-west direction. Similarly, *V* is reported at the northern and southern grid cell edges and centered in the east-west direction. To compute wind speeds in the grid box center, we average the 2 adjacent *U* and *V* entries and then calculate wind speeds. This approach means that one row and column at the fringe of the computational domain is undefined but this is not a problem because we remove the sponge layer anyway (see l. 117 for details).

A similar postprocessing was needed for the IDL simulations. In contrast to the GERICS simulations, this postprocessing was performed by the IDL modelers before sharing the data and it is thus not a part of the analysis presented in this study. Nevertheless, we explain the steps of the IDL postprocessing in the following:

> IDL geopotential is outputted on numbered levels *lev+1*, whilst wind is reported on numbered levels *lev'*. In particular, geopotential is given on a half vertical grid cell, and a linear interpolation of geopotential values from the *lev+1* to the *lev'* coordinate was performed. Winds are output on a staggered Arakawa C-grid scheme. That is, the U-component is given at the center of western and eastern edge of the grid cell, having the same number of points in the y-direction and one more point in x-direction. V-component is given at the center of northern and southern grid cell edges, having the same number of points in the x-direction and one more point in y-direction. This staggered Arakawa C-grid scheme was converted to an unstaggered Arakawa A-grid scheme and rotate the grid-relative winds to earth-relative winds, to compute the wind speeds. In this way, wind speeds are on the center of the grid cell and the geopotential are at the vertical midpoint.

Since the postprocessing was performed before the data was handed over, IDL (i.e., WRF) winds and geopotential are on the same vertical *mlev* coordinate and at the same rotated latitude and rotated longitudes. We therefore used the IDL data directly without similar postprocessing as described for the GERICS simulations above.

Following your comment, we decided to add an explanation of the GERICS postprocessing in the manuscript as we agree that it is crucial to document this part of the analysis better. We want to emphasize that the preprocessing is performed transparently in the github repository that we will publish upon publication. We provide a screenshot of the relevant code section below.

As a side note: We spent quite some time understanding the vertical dimension and we believe that your question is a perfect example of the difficulties that non-climate modelers face when using raw model output. This is why we argue that winds at different heights between 100m and 200m would be invaluable for usage in the impact modeling and wind energy community (see lines 327-332).

```
12  ######################
13  # Combine ground geopotential (FIB) with upper levels (FI) and interpolate from lev_2 to lev
14  ######################
15  for year in range(1986, 2016):
16      print(year)
17      for experiment in EXPERIMENTS:
18          # Load geopotential and drop non-needed variables
19          ds = xr.open_dataset(data_path + experiment + "/FI/FI_" + str(year) + ".nc")
20          ds = ds.drop(["hyai", "hybi", "hyam", "hybm"])
21          # Load surface geopotential and assign lev_2 coordinate (counted from top of atmoshere down)
22          ds_ground = xr.open_dataset(
23              data_path + experiment + "/FIB/FIB_" + str(year) + ".nc"
24          )  # Geopotential at surface
25          ds_ground = ds_ground.assign_coords({"lev_2": 28.0}).rename(
26              {"FIB": "FI"}
27          )  # ground is lowermost level
28          # Combine surface and further up
29          ds_combined = xr.concat([ds, ds_ground], dim="lev_2")
30          # Perform rolling mean interpolation (lev is defined between lev_2)
31          ds_combined = ds_combined.rolling({"lev_2": 2}).mean().dropna("lev_2")
32          # Rename variable from lev_2 to lev and align counting with other datasets
33          ds_combined = ds_combined.rename({"lev_2": "lev"}).assign_coords(
34              {"lev": [float(x) for x in range(1, 28)]}
35          )
36          # save
37          ds_combined.to_netcdf(
38              data_path + experiment + "/FI_interpolated/FI_" + str(year) + ".nc"
39          )
40
```

**Fig. 1:** Screenshot of the GERICS vertical preprocessing in *preprocess_GERICS.py*.

```
41  ######################
42  # Compute wind speeds from half-shifted wind components
43  ######################
44  for year in range(1986, 2016):
45      print(year)
46      for experiment in EXPERIMENTS:
47          # Open files
48          ds_u = xr.open_dataset(data_path + experiment + "/U/U_" + str(year) + ".nc")
49          ds_v = xr.open_dataset(data_path + experiment + "/V/V_" + str(year) + ".nc")
50
51          # u wind component at grid center is mean of u at its western and eastern margin
52          ds_u = ds_u.rolling({"rlon_2": 2}).mean()
53          ds_u = ds_u.rename({"rlon_2": "rlon"}).assign_coords({"rlon": ds_v.rlon})
54          # same for v with northern & southern margin
55          ds_v = ds_v.rolling({"rlat_2": 2}).mean()
56          ds_v = ds_v.rename({"rlat_2": "rlat"}).assign_coords({"rlat": ds_u.rlat})
57
58          # Compute wind speeds from components
59          ds = xr.merge([ds_u, ds_v])
60          ds["S"] = (ds.U**2 + ds.V**2)**(1./2)
61          ds.S.attrs = {"long_name": "Wind speed", "units": "m/s", "grid_mapping": "rotated_pole"}
62
63          # drop non-needed vars and save
64          ds.drop(["U", "V"]).to_netcdf(data_path + experiment + "/S/S_" + str(year) + ".nc")
```

**Fig. 2:** Screenshot of the GERICS horizontal preprocessing in *preprocess_GERICS.py*.

**Changes to the manuscript**

l. 127

**2.4 Preprocessing the horizontal and vertical coordinates**

GERICS reports wind components *u* and *v* at the horizontal grid box edges and we interpolate them linearly to obtain wind components at the grid box centers. Moreover, geopotential and horizontal wind components are reported at different vertical levels which are separated by half a grid box in the vertical. After consultation with the modelers, we decided to interpolate the geopotential linearly to the vertical position of the winds. These two steps allow us to compute wind speeds at the same horizontal locations as in the IDL simulations and with known elevation above ground. The implementation is documented in preprocess_GERICS.py which is part of the github repository belonging to this publication (see code availability statement).

The IDL model output was already postprocessed to provide wind components and geopotential height at the same vertical levels and horizontal coordinates before the data was handed over to us. We therefore did not perform a postprocessing as outlined above but used the data as provided.

**2.45 Comparison to the log and power laws …**

**Reviewer comment 2-5**

*Is only the lowest model level influenced by the change to all forest in both models? Presumably the trees could physically extend to more than ~30m tall and therefore into the 2nd lowest model level but the parameterizations and models prevent such complex dynamics?*

**Author response**

That's a good point. Some very big trees would indeed peak into the second model level if they were modeled explicitly. However, as you correctly presume, vegetation is not physically resolved but parameterized in the GERICS and IDL models because the resolution is substantially coarser than the vegetation features. As an illustration: a single grid box in the lowermost level has a volume of order of 100 km$^3$ (50 km x 50 km x 30m) which is about 10$^8$ times larger than the volume of a large tree (5m x 5m x 30m = 750m$^3 \sim$ 10$^{-6}$ km$^3$). The models therefore use simplified relationship to capture the effects of vegetation on surface climate. GERICS, for instance, employs a bulk land surface model, which treats the impact of surface obstacles such as trees on the momentum flux through the roughness length. In such bulk land surface models, the land surface is generally considered as having infinitesimal vertical extent as discussed in greater detail in Breil et al. (2020). To flag limitations of current regional climate models, we now explicitly mention that vegetation is only parameterized in the manuscript, see below.

**Changes to the manuscript**

l. 114 f.

Vegetation is parameterized in the IDL and GERICS models and it impacts the atmosphere via changes in surface parameters like roughness length and albedo. Parameterization is needed because explicit physical modeling of individual trees would require substantially finer grid resolution than the 50km grid spacing available from LUCAS. One implication of the parameterization and large grid box sizes is that trees can not directly impact atmospheric flow in higher atmospheric levels even if very tall real trees exceed 30m and would thus reach into the second model level. As shown in this manuscript, however, trees do impact atmospheric flow further up via surface changes that are mediated upwards.

2.3 Area and period of interest

**Reviewer comment 2-6**

*Interpolation from hub-height wind speeds from both models from the modeled wind speeds above and below hub-height*

*Where is it described how the extrapolation approaches (Eq. 2, Eq. 3) are not enough according to your title?*

*I expected to see comparisons of these two approaches to the interpolated model levels, but where are these comparisons?*

**Author response**

Below we provide a list of citations from our manuscript that describe how the extrapolation approaches are not good enough:

l. 255-257
"Overall, we conclude that changes in the wind profile caused by afforestation/deforestation are highly complex. Constant extrapolation from the surface values would miss essential features of changes in the daily cycle, and is therefore not well suited to compute sub-daily hub height wind speeds needed for wind power conversion."

l. 285-287
"While we can not robustly pinpoint a single explanation for the jet occurrence, our results emphasize the need for explicit modeling rather than extrapolation of surface winds, as low-level jets would generally not be captured using the log and power laws."

l. 322 – 326
"The reported changes are complex, non-local, depend on the location, impact the daily and seasonal cycle, and are therefore not well captured by simple heuristics like the log law and

power law even when using modified parameters. Using the log and power law in simulations with substantially different land use with the same parameters is particularly detrimental because it overestimates the effects of land use change on hub height winds, and might mask dynamical changes. We therefore strongly recommend to not scale 10m climate model output to hub heights with constant coefficients in the log or power law."

**Reviewer comment 2-7**

*2 includes the displacement height (d) to account for turbulence differences with the atmospheric surface layer but all that seems to be changing is the roughness length (line 146)*

*Are you suggesting that displacement height doesn't need to change in a fully forested onshore scenario poised to estimate wind speeds?*

**Response to the reviewers**

No, we are not suggesting that displacement height does not need to change. It has to change when grass is replaced by forest. Our argument is more subtle and relates to the state-of-the-art in climate change assessments for wind energy. Since the land-use change is only one component in the climate change scenarios (e.g., representative concentration pathways or Shared Socioeconomic Pathways) and receives much less attention than greenhouse gas emissions or radiative forcing (which even features in the scenario names!), land-use change is typically ignored in the scientific literature on future wind energy potentials. We challenge this status quo that uses the same wind profile in the future and the past (i.e., the same power law coefficient or the same roughness length and the same displacement height). The point of the current study is to show that the status quo creates misleading results because it scales the effect of land-use change incorrectly.

Some examples of the status quo, summarizing the methodology applied in the papers that we cite in the Introduction. Please note that the list includes our own work: we argue that we need to improve collectively as a scientific community of practice.

1. **Hueging et al., 2013** : They use 2 regional climate models to analyze climate change impacts on wind energy in the 21$^{st}$ century. They use the power law with "power-law exponents a of 0.2 for onshore areas (IEC 2005a) and of 0.14 for offshore sites (IEC 2005b)" to extrapolate from 10m to hub height. That is, they use the same wind profile in the future.
2. **Tobin et al., 2016** : They use EURO-CORDEX regional climate models to analyze climate change impacts on wind energy in the 21st century. They extrapolate 10m wind speeds to hub height using the power law with a fixed coefficient of 1/7 (see their supplementary material, page 10). That is, they use the same wind profile in the future.
3. **Reyers et al., 2016** : They use CMIP5 simulations and statistical-dynamical downscaling to analyze climate change impacts on wind energy in the 21$^{st}$ century. They use the power law to extrapolate from 10m to 80m with a constant power law coefficient with the same values as in Hueging et al., 2013 (see Reyers et al., 2015 for details about the method). That is, they use the same wind profile in the future.

4. **Karnauskas et al., 2018** : They use CMIP5 simulations to evaluate global wind energy potential and how it develops under climate change. Their method: "The 10-m wind speed fields are extrapolated to 100 m using a power law with coefficient 1/7". That is, they use the same wind profile in the future.

5. **Schlott et al., 2018**; They use EURO-CORDEX simulations together with PyPSA power system modeling to quantify the effect of climate change on the European power system. They use the log law with a roughness length "which is provided by the datasets as a static quantity" and a displacement height of zero to extrapolate from 10m to 90m hub height. That is, they use the same wind profile in the future.

6. **Soares et al., 2019**: They use regional climate model simulations (largely from CORDEX Africa) to evaluate the effect of climate change on wind energy resources in Northwestern Africa. For the CORDEX-Africa simulations which only provide 10m winds, they use the power law to extrapolate from 10m to 100m and 250m. That is, they use the same wind profile in the future (at least for parts of the analysis).

7. **Lima et al., 2021**: They use the same approach as in Soares et al., 2019 to study the present and future wind resource in South-Western Africa. That is, they use the same wind profile in the future (at least for parts of the analysis).

8. **Wohland et al., 2021**: They use EURO-CORDEX to study the effect of climate change on wind energy complementarity in Europe. They use the power law with a fixed coefficient of 1/7 to extrapolate from 10m to 80m hub height. That is, they use the same wind profile in the future.

9. **Bloomfield et al., 2020:** They use reanalysis and selected EURO-CORDEX simulations (i.e., the ECEM dataset) to quantify the effect of climate change on different types of renewable generation, including wind energy. Their assessment is based on 10m winds and they extrapolate to 100m using the power law with a fixed 1/7 exponent. That is, they use the same wind profile in the future.

All studies listed above use the same wind profile in the future even though land-use is poised to change in the analyzed scenarios. We hope that this list helps to clarify the context of our paper and how we aim to contribute to better climate-model based wind energy assessments with it.

We understand that your concern is partially also related to Fig. 3. We would like to emphasize that the mean match between the adapted extrapolation and the model output (as shown in Fig. 3) could be further improved by fine-tuning the displacement height. Please note that we already conclude that "the results show that the decay … is fairly linear in log space and can thus be approximated well by the log and power law on average over all seasons, 30 years, and an entire continent" (l. 213), and showing that the average agreement could be even better with parameter fine tuning does not create additional insights. Moreover, such fine tuning is not relevant for the key findings of this study because the issue with the log and power law is more fundamental. Displacement height changes could only substantially contribute to fixing the mismatch between extrapolation from 10m and interpolation from model levels if they were a function of the hour of the day and the season. While seasonal variations are generally plausible for deciduous trees, a daily cycle in displacement height is not plausible.

While reflecting on your comment, we realized that we did not consistently report our choice of displacement height which we do now, see below. We also identified a small inconsistency in our parameter choice. While the displacement height was set to d=3m for the common extrapolation (explained in l. 196), the displacement height for the adapted extrapolation was set to d=0m for both the grass and the forest profile. As agreement with the model output deteriorates slightly when using d=3m, we now use d=0m in all cases. This alternative choice has negligible impacts on Fig. 3 (we provide the old and new version below) and no further implications beyond Fig. 3.

**New Fig. 3 (with d=0m in all log profiles)**

[Figure]

**Old Fig. 3 (with d=3m for green low law curve and d=0m for the others)**

[Figure]

**Changes to the manuscript**

l. 143

When comparing model level results with extrapolations (in Fig. 3), we use the widely adopted power law exponent alpha=1/7. To compute the log-law profiles, we use roughness lengths as reported in Breil et al. (2020) for IDL, assuming a 50/50 split between coniferous and deciduous treses. The GERICS values from that paper, however, can not be directly used because the contribution from subgrid-scale orography is missing. We therefore computed onshore mean effective roughness lengths from the climate model (FOREST: zo = 1.686m, GRASS: zo = 0.693m) and use those values. We set displacement height d=0m in the log-law plots in Fig. 3.

l. 195

with the same profile in the GRASS and FOREST scenarios (i.e., following Eq. 2 and 3 with constant $\alpha$ = 1/7, z0 = 0.05 m corresponding to grass, and d = 30 m).

**Reviewer comment 2-8**

*Supplement includes extra figures but not a more detailed methodology – manuscript suggests the python code will be made available on github upon publication*

**Author response**

Yes, both is correct. As mentioned before, we are happy to share the code with you if you want to have a look now. It's on github and we could create a test profile for you that would allow you to access it without giving up your anonymity. We also invite you to have a look at the github repository that we shared with our last Earth System Dynamics publication: https://github.com/jwohland/stilling_MPI-GE. We believe that this example illustrates our dedication to foster reproducibility.

**Reviewer comment 2-9**

*Results: Section 3.2 (Wind change profiles per season at individual locations)*

*Here I was expecting to learn when the extrapolation estimates from either surface (10m) variables or model levels struggled, but it is a comparison between IDL and GERICS*

**Author response**

We hope that our explanation to your comment 2-7 helped to clarify the issue. Please let us know if it doesn't.

**Reviewer comment 2-10**

*Section 3.4: Low level jet– agree that surface wind speeds are not appropriate for estimating the LLJ speed or simply the presence of an LLJ (saying nothing about an LLJ being defined by a 'nose' that requires slower wind speeds above it and below it; not possible from either wind speed extrapolation approach) – I don't understand why a discussion about LLJ is in here as a text bridge to how the LLJ may empower or damage wind turbines isn't made*

**Author response**

Thanks for another very good comment. We do not investigate damage at all in this study even though we agree that it is important. However, it is beyond the scope of the current study. We add a brief pointer at the end of the manuscript to flag turbine damage as an interesting field for future work.

Unfortunately, we don't fully understand your last sentence (*I don't understand why a discussion about LLJ is in here as a text bridge to how the LLJ may empower or damage wind turbines isn't made).* We *assume* that you refer to the short paragraph in lines 264-267. We added this paragraph to introduce an LLJ as a possible explanation of why winds may be higher at nighttime in summer over a forested Central Europe and to answer the question "why would winds be stronger over forest than over grass". Please see our response to **comment 1-7** from the other review for more detail. Please let us know if we misinterpreted this part of your comment.

**Changes to the manuscript**

> See changes presented in the next comment as both changes apply to the same part of the manuscript.

**Reviewer comment 2-11**

*Capacity factor plots: Given the enormous research and economic interests in quantifying wake effects, maps of capacity factors (such as Fig. 7a,b,d,e) can easily be mis-interpreted that what is actually being shown is that turbine's power curve being applied to the winds without including a between-turbine spacing consideration or wake effect consideration – realilzed capacity factors will be much lower even if the wind speeds in the model are perfect everywhere and all the time (and assuming Europe is all grass or all forest)*

**Author response**

Thanks. We agree that turbine-turbine and park-park interactions matter when it comes to credible yield estimation. However, both are beyond the scope of this study and would require different modeling approaches to tackle. We explain explicitly in the Introduction that we aim to isolate the effect of land-use change on wind profiles (see lines 46 ff). We add a sentence to clarify that we ignore wakes, blockage and the like (see below).

Moreover, we would like to point out that there are cases where turbine-turbine and park-park interactions do not matter, namely when installed capacities are low. In any case, we do not make assumptions about the deployment and siting of future wind parks either.

We believe that we should be humble in communicating the limits of our work. And we hope that the framing in the Introduction together with the additions outlined below make it clear where the presented study improves relative to the state-of-the art and where complexity is reduced on purpose in a justifiable manner.

**Changes to the manuscript**

l. 161

> represents the median current wind turbine. Please note that we ignore turbine-to-turbine and park-to-park interactions like wake effects, wind farm blockage and resource depletion by upstream wind parks in this study. We also do not model turbine damage and ageing caused by changes in the wind resource, such as changes in gusts. While we acknowledge that these effect matter for wind energy yield assessments, capturing them is beyond the scope of the current study and would require a different and more highly resolved model setup.

---

## Author Response (AR2)

**Response to the reviewers**

**Extrapolation is not enough: Impacts of extreme land-use change on wind profiles and wind energy according to regional climate models**

Jan Wohland, Peter Hoffmann, Daniela C. A. Lima, Marcus Breil, Olivier Asselin, and Diana Rechid

Under review for ESD

Status: Publish subject to minor revisions after first round of revisions

We would like to thank both reviewers for their clear assessments of our submission. We are glad that reviewer #1 is satisfied with our first round of revisions and that they now recommend *publish as is*. We understand that reviewer #2 has not taken part in the first round of revisions and that they have additional good comments which we address below.

Throughout this document, we use *italics* to mark the reviewer comments,  to mark text that we deleted from the manuscript and blue to mark text additions to the manuscript. When reporting line numbers, we refer to the clean manuscript in the first round of revisions (i.e., the most recent version before this revision).

**Report #1**

*Wohland et al's efforts to fix various parts of this study are done to my satisfaction. I also appreciate they utilized this chance to expand upon a few technical steps as well as highlight why the research is interesting. Thank you.*

**Author response**

Thanks for taking the time to evaluate our first response. We are glad that you consider our modifications to be satisfactory.

**Report #2**

*The study presents the effect of extreme land use change on wind energy sources. Specifically, they evaluated the affect of afforestation/deforestation on wind energy. It uses model outputs from REMO-iMOVE run by GERICS and WRFaNoahMP run by IDL, which are products of LUCAS consortium. The authors have computed mean wind speeds for a time period of 1986 to end of 2015. It considers changes in annual mean as well as seasonal and daily cycle.*

*There are two major purposes of this study. First one is to evaluate the methodologies that are used for evaluating hub-height wind speeds in the field of wind energy: a) the conventional one that extrapolates using power law log-law from surface winds, or b) interpolating winds from model outputs, such as WRF and REMO-iMOVE. Second is to evaluate a possible effect of afforestation on future wind speed projection and hence, wind power. The results from the study can be helpful in better management and forecasting of wind energy resources in future.*

*The study is mostly well presented and thought. The authors have compared a conventional method of extrapolating wind speeds, with the output from dynamical models. They have found limitations in it from the perspective of climate change (afforestation).*

**Author response**

We are glad that reviewer 2 considers our evaluation to be well presented and thoughtful.

**Comment 2-1**

*I have a few questions from the author: I understand that the power law used here in this study are valid for a neutral atmospheric condition. In that case will it be more relevant to find only the instances from the model output where the atmosphere is neutral for an apple-to-apple comparison. Kindly address what can be the affect of atmospheric stability in using these techniques in section 3.1. Please check for log law as well.*

**Author response**

Thanks for your comment. Yes, we agree that the power law can be a useful heuristic under certain atmospheric conditions. And we also acknowledge that the log law can even be formally derived from the atmospheric equations of motion as a special case (l. 156: "The log law can be formally derived as a special case of the general equations of motion under neutral stratification, while the power law is empirically motivated (e.g., Emeis, 2013)".

Our goal is not so much to verify that the power law is valid when it should be (i.e. in the special case of a neutral atmosphere), but instead to show that the typical use of the power and log laws to extrapolate future climate model output is inappropriate. Indeed, even though those heuristics are sometimes useful, they are used under all atmospheric conditions in the published literature. Since we are interested in contrasting explicit modeling with the state-of-the-art in climate-model based wind energy assessments, the proper apple-to-apple comparison to compare to the extrapolations applied at all times. Below you find a list of examples that documents that log or power laws with constant (!) coefficients are typically used to extrapolate near-surface winds to hub height. The list is copied from our first response to reviewers, assuming that you did not have access to it. Please apologize for the duplicate should you have had access to it already:

1. **"Hueging et al., 2013** : They use 2 regional climate models to analyze climate change impacts on wind energy in the 21$^{st}$ century. They use the power law with "power-law exponents a of 0.2 for onshore areas (IEC 2005a) and of 0.14 for offshore sites (IEC 2005b)" to extrapolate from 10m to hub height. That is, they use the same wind profile in the future.
2. **Tobin et al., 2016** : They use EURO-CORDEX regional climate models to analyze climate change impacts on wind energy in the 21st century. They extrapolate 10m wind speeds to hub height using the power law with a fixed coefficient of 1/7 (see their supplementary material, page 10). That is, they use the same wind profile in the future.
3. **Reyers et al., 2016** : They use CMIP5 simulations and statistical-dynamical downscaling to analyze climate change impacts on wind energy in the 21$^{st}$ century. They use the power law to extrapolate from 10m to 80m with a constant power law coefficient with the same values as in Hueging et al., 2013 (see Reyers

et al., 2015 for details about the method). That is, they use the same wind profile in the future.

4. **Karnauskas et al., 2018** : They use CMIP5 simulations to evaluate global wind energy potential and how it develops under climate change. Their method: "The 10-m wind speed fields are extrapolated to 100 m using a power law with coefficient 1/7". That is, they use the same wind profile in the future.

5. **Schlott et al., 2018**; They use EURO-CORDEX simulations together with PyPSA power system modeling to quantify the effect of climate change on the European power system. They use the log law with a roughness length "which is provided by the datasets as a static quantity" and a displacement height of zero to extrapolate from 10m to 90m hub height. That is, they use the same wind profile in the future.

6. **Soares et al., 2019**: They use regional climate model simulations (largely from CORDEX Africa) to evaluate the effect of climate change on wind energy resources in Northwestern Africa. For the CORDEX-Africa simulations which only provide 10m winds, they use the power law to extrapolate from 10m to 100m and 250m. That is, they use the same wind profile in the future (at least for parts of the analysis).

7. **Lima et al., 2021**: They use the same approach as in Soares et al., 2019 to study the present and future wind resource in South-Western Africa. That is, they use the same wind profile in the future (at least for parts of the analysis).

8. **Wohland et al., 2021**: They use EURO-CORDEX to study the effect of climate change on wind energy complementarity in Europe. They use the power law with a fixed coefficient of 1/7 to extrapolate from 10m to 80m hub height. That is, they use the same wind profile in the future.

9. **Bloomfield et al., 2020:** They use reanalysis and selected EURO-CORDEX simulations (i.e., the ECEM dataset) to quantify the effect of climate change on different types of renewable generation, including wind energy. Their assessment is based on 10m winds and they extrapolate to 100m using the power law with a fixed 1/7 exponent. That is, they use the same wind profile in the future.

All studies listed above use the same wind profile in the future even though land-use is poised to change in the analyzed scenarios. We hope that this list helps to clarify the context of our paper and how we aim to contribute to better climate-model based wind energy assessments with it." (Copied from first Author response).

**Comment 2-2**

*Title of the paper is quite strong: "Extrapolation is not enough". My question is: Is there any possibility that only extrapolation, may be along with calibration based on different scenarios can work?" I also request authors to kindly consider rephrasing the title.*

**Author response**

Thanks for this comment. As we show throughout the paper, wind profiles vary, for example, with time of day, season, and lower boundary forcing. As you rightly point out in your first comment, these variations are partly related to atmospheric stability, and hence physically expected.

The fundamental problem with the log and power law (see eq. 2 & 3 in the manuscript) is that the profiles themselves are not temporally dependent (see lines 137 to 142 in manuscript) and the parameters *alpha* and *z0* and *h* are typically not even changed in climate projections with strong land-use change. That means that the log and power law are unable to reproduce the variations with time of day, season and lower boundary forcing that we report based on model-level output. So, to answer your question: it is not possible that extrapolation can "work" as long as it doesn't explicitly include the spatio-temporal complexity of wind profile changes. We therefore decided to keep the title as is after considering your suggestion.

Please note, however, that our suggested method (i.e., interpolation between model levels) takes the form of a power law with the essential modification that *alpha* is not longer a single number but a function of time and space, also varying per scenario. To compute this version of *alpha,* one needs knowledge about wind speed above and below the height of interest. We do not think that it is justified to label this version of the power law as an extrapolation because it is an interpolation.

**Comment 2-3**

*Line 268, Section 3.2: Why in "summer regions there is a drop and then in winters there is an increase in wind profile change?"*

**Author response**

Unfortunately, we could not find the sentence that you quote above in our manuscript.

The paragraph around line 268 in the clean version of the manuscript discusses Fig. 4 and states that "surface changes are very high in winter (ca. 3m/s), high in the annual mean and spring (ca. 2 m/s), medium in fall (ca. 1.75 m/s), and comparably low in summer (ca. 1.2m/s). By contrast, wind speed change is approximately 0.8 m/s at the highest displayed level (around 750m) in all seasons."

In the tracked changes version of the manuscript, we write in line 268: "The models also agree that wind speed changes decay monotonically with height although exceptions exist during the summer, where IDL projects a local minimum followed by increased change (Fig. 4b,c)."

Maybe you could clarify what your question exactly refers to and whether the text in quotation marks ("") is an actual quote or something else? Thanks.

**Comment 2-4**

*Section 3.3: L289 The surface perturbation decays slowly around noon and decays quicker in night. There is more mixing in atmosphere around noon, yet the perturbation because of introduction of a forest decays slowly. I request authors to consider adding*

*justification.*

**Author response**

Thanks for this comment. If we understand correctly, you are challenging the results presented in Fig. 5 (2nd and 4th column) because you expect wind profile changes to decay more quickly around noon because there is more mixing. Actually, the opposite is true: more mixing means that the surface change also manifests further aloft (because the near-surface wind speed reduction is not constrained to the surface but "spreads" to upper levels via mixing).

Let's take April as an example. Panels m and o in Fig. 5 show that low wind speed values do not only occur near the surface but also aloft in the GRASS simulations, and this effect is most pronounced at 12h. In other words, there is indeed more mixing at 12h in the GRASS simulations, as expected since downwelling solar radiation peaks around noon. At the same time, panels f and h show that the GRASS-FOREST difference has the largest vertical extent at noon as well. That is, the reduction near the surface also manifests at the models levels further aloft. Or in other words:

> "In particular, we find that the surface perturbation decays slowly around noon and decays relatively quickly at night during all months in IDL (Fig. 5, panels d, h, l, and p) and during all months but October in GERICS (Fig. 5, panels b, f, j, and n)." (Quotation from the manuscript, l. 275)

We hope that these explanations helped to resolve the confusion. We decided to add the following short explanation to the manuscript to avoid similar confusion among the readers.

**Changes to the manuscript**

l. 279:

> during all months but October in GERICS (Fig. 5, panels b, f, j, and n). This daily cycle is consistent with physical expectation: increased mixing around noon implies that the near-surface wind reductions impact higher level winds more strongly. Similarly, more stable nighttime conditions imply that the near-surface wind reductions are more constrained to the surface. While models...

**Comment 2-5**

*Please recheck if all the supplementary figures are discussed in the main paper*

**Author response**

Thanks for spotting this one. There was indeed a problem that was caused by an earlier reformatting of the SI which meant that SI Figures S2 – S9 were not properly referenced. We changed the text to fix this issue, see below. We made sure that all SI Figures are now referenced in the manuscript.

**Changes to the manuscript**

To test the robustness of change in different parts of the year, we analyzed the changes per season (see Supplementary  Figs. S2 – S9).

**Comment 2-6**

*Why are there two peaks in Figure 8 IDL CF?*

**Author response**

We are not aware of any a priori reason to expect one, two, or more peaks in the distribution of capacity factors. The bimodal shape of the IDL distributions implies that CF of around 0.4 or around 0.55 in the GRASS simulation occur more often than other values. By contrast, the GERICS distributions feature a single peak. Because of this finding, and a few others, we flag repeatedly that there is considerable model uncertainty, for example:

- l. 381: "Since we also report substantial model uncertainty, the exact values provided here should be treated with caution"
- l. 264 "In other words, model uncertainty is high at individual locations."
- l. 212 "Additionally, disagreement can stem from the allocation of land surface surface parameters (e.g., roughness length and leaf area index) and whether and how those parameters evolve throughout the year."

Since other readers might also wonder why the distributions are qualitatively different, we add a brief discussion (see below). Thanks.

**Changes to the manuscript**

l.352:

Moreover, while GERICS features a single distinct peak in both experiments, IDL is weakly bimodal. The single peak in the GERICS GRASS distribution means that values around 0.3 occur most frequently. By contrast, the bimodal shape of the IDL distributions implies that CF of around 0.4 or around 0.55 in the GRASS simulation occur more often than other values. While we are not aware of any a priori reason to expect one, two, or more peaks in the distribution of capacity factors, this qualitative difference in shape is another example of model uncertainty.